# KNOWLEDGE DISTILLATION THROUGH GEOMETRY-AWARE REPRESENTATIONAL ALIGNMENT

## ABSTRACT

Knowledge distillation is a common paradigm for transferring capabilities from larger models to smaller ones. While traditional distillation methods leverage a probabilistic divergence over the output of the teacher and student models, feature-based distillation methods often minimize variants of Euclidean norms between the hidden layer representations. The main goal is for the student to mimic the structure of the feature space of the teacher. In this work, we theoretically show that existing feature distillation methods, such as projection based mean squared loss or Centered Kernel Alignment (CKA), cannot capture the feature structure, even under zero loss. We then motivate the use of *Procrustes distance* and the Frobenius norm of *Feature Gram Matrix*, distances already common in the context of measuring representational alignment, as distillation losses. We show that feature distillation through our method showcases statistically significant improvement in distillation performance across language models families (BERT and OPT) in classification and instruction-following tasks by up to 2 percentage points, showcasing the potential of integrating feature geometry into existing distillation methods. [1]

## 1 INTRODUCTION

While large models are achieving state-of-the-art results across almost all vision and language tasks, the emergent abilities these models exhibit (Wei et al., 2022; Liang et al., 2023b) are often inaccessible to the public as a result of their inherent size and operating costs. Knowledge Distillation (KD) is one of the many paradigms that aim to bridge the gap between size and performance by inducing ways of transferring knowledge and abilities from a larger, complex model (teacher) to a smaller and accessible model (student).

Assuming white-box access (weights and intermediate representations) to the teacher model during the training process, we can leverage alignment of the teacher-student model through not just their outputs, but also their hidden representations. (Sanh et al., 2020; Liang et al., 2023a; Sun et al., 2019; Mukherjee & Hassan Awadallah, 2020). These methods, often called *feature distillation*, construct a loss function that quantifies the informational gap between the teacher and student model representations.

A longstanding challenge in feature distillation is the dimension mismatch between the student and teacher representations. The standard approach mitigates this issue by learning a linear projection from the student's representation space to the teacher's, enabling the application of simple similarity measures such as the Euclidean distance (Jiao et al., 2020). More recent work on feature distillation (Dasgupta & Cohn, 2025) has used Centered Kernel Alignment (CKA) (Kornblith et al., 2019), a kernel based measure originally introduced to compute the (dis)similarity between deep learning models. CKA operates on the Gram matrix between features, and is thus agnostic to the dimension mismatch problem. CKA comes from a wider literature in representational alignment (Sucholutsky et al., 2023), where various other functions for comparing the similarity of neural networks have been proposed. (Klabunde et al., 2023). We propose using Procrustes distance (Schönemann, 1966; Williams et al., 2021) and the Frobenius norm of the Gram matrix differences (Yin & Shen, 2018), alternative methods that have been proposed in the representational alignment literature. We justify

---

[1] `https://github.com/x-labs-xyz/feature-distillation`

their use in feature distillation through a theoretical framework, demonstrating that they more faithfully capture the geometric alignment of feature representations compared to CKA and projection-based methods.

While the representations generated by language models can vary based on a myriad of factors (Lampinen et al., 2024), it has been noticed that relative representations (angles and inner products) are preserved for models trained on the same task with the same data. (Moschella et al., 2022). Thus, our definition of *feature geometry* is equivalent to that of a spherical geometry on a unit normed sphere. We question the hypothesis that task-specific feature distillation is correlated with the preservation of this feature geometry between the student and teacher models.To rigorously assess this hypothesis, we conduct a theoretical examination of prevalent feature distillation objectives, complemented by empirical studies on their effectiveness in task-specific language model distillation.

Our core contributions are summarized below:

- We show, theoretically and through a synthetic experiment, that optimizing over CKA and linear projection does not always correlate with the preservation of geometry in feature representations. In contrast, we show that Procrustes distance is a better proxy for feature geometry alignment.
- We show that Procrustes distance outperforms CKA and other feature distillation baselines on classification tasks using BERT.
- We show that optimizing over Procrustes and the Frobenius norm of the difference between Feature Gram matrices outperforms CKA in instruction-following task using OPT.

## 2 BACKGROUND

### 2.1 KNOWLEDGE DISTILLATION

The distillation process is usually done by gradient descent on a loss that minimizes the student target loss, as well as a secondary loss that incorporates the difference in the "knowledge" being transferred from the teacher to student model. Specifically, it takes the form of $\mathcal{L} = \mathcal{L}_{\text{CE}}(f_S(x), y) + \mathcal{L}_{KD}(f_T(x), f_S(x))$ where $f_S(x)$ and $f_T(x)$ are last-layer logits of the student and teacher model respectively, $y$ is the true output labels, $\mathcal{L}_{KD}$ is the KL divergence between teacher and student logits and $\mathcal{L}_{CE}$ is the cross entropy of the student output.

Traditional knowledge-distillation methods have used either the forward (Sanh et al., 2020; Hinton et al., 2015) or reverse (Agarwal et al., 2024; Gu et al., 2024) KL divergence as the measure of difference between the output logits. The large vocabulary size of modern language models means that minimizing probabilistic divergences over them can often lead to undesirable behaviors. In particular, minimizing the KL divergence leads to "mode-covering" behavior, where the student model is forced to exactly match the distribution of the parent model throughout its domain. As a result, the student model must spread probability mass across many tokens, including those that the teacher itself treats as low-likelihood. Reverse KL divergence attempts to solve this by focusing on tokens with high probability. However, this can easily lead to a lack of diversity in the generated output. Integrating information from intermediate representations can help alleviate some of these problems, resulting in the application of feature distillation

For feature distillation, it is natural to assume that $\mathcal{L}_{KD}$ can take the form of any vector $p$-norm. Variants of Euclidean norms, including cosine-similarity (Sanh et al., 2020), normalized mean-square, (Liang et al., 2023a; Sun et al., 2019) and $\ell^2$ norms (Mukherjee & Hassan Awadallah, 2020) have been used. A variety of higher order projection methods on Euclidean spaces can be used to bridge the dimension mismatch problem. However, the necessity to learn a linear projection is a significant drawback. Similarly, learned linear projections and Euclidean distances might be too powerful to reflect the geometry of neural representational spaces, which are invariant to permutations or orthogonality in the space of representations. (Kornblith et al., 2019; Rombach et al., 2020).

More recently, Centered Kernel Alignment (CKA) has been proposed instead of projection based methods for distillation of language models (Dasgupta & Cohn, 2025). CKA works on the Gram matrix of feature representations, thus avoiding the necessity to learn any additional projection meth-

ods. CKA is also endowed with useful properties such as invariances to orthogonal transformations and isotropic scalings, which reflect the symmetries of the representational space. Dasgupta & Cohn (2025) show that CKA does better than learned projection across model sizes and tasks. The distinction between projection-based and alignment-based feature distillation losses remains largely unexplored beyond end-to-end empirical comparisons.

## 2.2 Feature geometry of language models

With the increasing size and complexity of language models, a significant amount of work has been put into understanding the mechanisms through which these models perform complex tasks. A particularly influential line of work focuses on the geometric properties of these representations. The Linear Representation Hypothesis (LRH) (Elhage et al., 2022; Park et al., 2024) is motivated by empirical evidence of the linear separability of complex ideas such as gender (Bolukbasi et al., 2016), truthfulness (Li et al., 2023; Marks & Tegmark, 2023), and refusal (Arditi et al., 2024; Jain et al., 2024) in the representational space of language models. The LRH hypothesizes that a language model implicitly constructs a subspace within a unit sphere of the dimension size of the representations, with each semantically unrelated concept approximately orthogonal to each other. (Jiang et al., 2024). The fact that these models can represent more concepts than their dimensions is justified by the relaxation from exact orthogonality to approximate orthogonality; while there

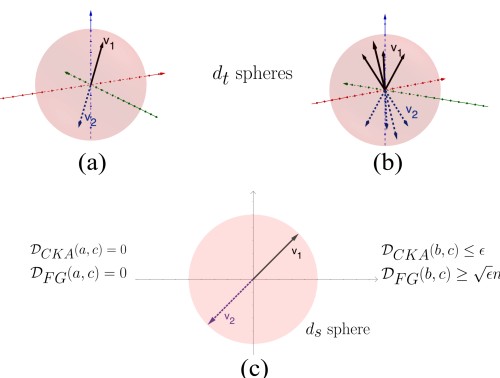

Figure 1: A simplified illustration of the phenomenon prescribed by Theorem 1. (a): $n$ vectors in $d_t$ dimensions lie in exactly two configurations that are antiparallel to each other. (b): A subset of those $n$ vectors from (a) are perturbed along a distinct orthogonal direction among the $d_t$ possible ones. (c): an exact replication of (a) in $d_s < d_t$ dimensions. Although the feature geometries differ, CKA computed with respect to (c) fails to differentiate between (a) and (b).

can only be $d$ vectors that are exactly orthogonal to each other in $\mathbb{R}^d$, the Johnson–Lindenstrauss lemma (Johnson et al., 1984; Dasgupta & Gupta, 2003) guarantees that there at at least $2^{O(\epsilon^2 d)}$ vectors whose pairwise inner product is less than $\epsilon$. These empirical insights demonstrate the role of feature geometry in how language models structure and encode knowledge. While large models, as a result of additional dimensionality, are naturally more capable of developing such structured representations during pretraining, smaller models trained in isolation can fail to replicate this structure. By explicitly guiding the student model's representations to align with the teacher's inner product structure, feature distillation offers a direct mechanism for preserving this geometric structure.

Besides construction through the LRH, the inner product structure between the latent features of language models has also demonstrated unique properties. While the learned feature representation of language models have been shown to be biased by task, complexity and learning order, (Lampinen et al., 2024), the angles between latent embeddings of models trained under the same data have been observed to be preserved under the same data and modeling choices (Moschella et al., 2022). This geometric invariance has been exploited for tasks such as latent state communication (Maiorca et al., 2023) and universal translation (Jha et al., 2025), motivating further investigation into its impact on the effectiveness of distillation.

## 2.3 Geometry preserving feature distillation in vision models

Similar geometric methods, while not always articulated and motivated from the same lens, have seen broad application for feature distillation in vision models. Differences in Gram matrix norms have been widely used, either explicitly or implicitly, in feature distillation loss functions, consis-

tently yielding performance gains (Tung & Mori, 2019; Park et al., 2019; Tian et al., 2020; Miles et al., 2024; Mannix et al., 2024). Distilling the geometrical neural collapse structure (Papyan et al., 2020) in image classification models has also shown empirical gains (Zhang et al., 2025). A rigorous theoretical foundation for these methods remains underdeveloped, and their application to language models has largely been confined to architecture or task-specific adaptations.

## 3 METHODS

### NOTATION

Consider the case where there are $n$ features, each with a deterministic feature direction. Let $\mathbf{R_t} = [\mathbf{v_1} \dots \mathbf{v_n}]^T \in \mathbb{R}^{n \times d_t}$ and $\mathbf{R_s} = [\mathbf{w_1} \dots \mathbf{w_n}]^T \in \mathbb{R}^{n \times d_s}$ be the matrices of unit norm representations of the $n$ features from the teacher and student network respectively, such that $\mathbf{v_1} \dots \mathbf{v_n} \in \mathbb{R}^{d_t}$ whereas $\mathbf{w_1} \dots \mathbf{w_n} \in \mathbb{R}^{d_s}$ and $d_t > d_s$. We will use $\|X\|_F$ to represent the Frobenius norm of the matrix $X$(square root of the sum of squared singular values) and $\|X\|_*$ to denote the nuclear norm of $X$ (the sum of singular values). Let $\mathbf{K_t} = \mathbf{R_t R_t^T}$ and $\mathbf{K_s} = \mathbf{R_s R_s^T}$ be $\mathbb{R}^{n \times n}$ Gram matrix of the teacher and student features respectively. We denote the all-ones matrix in $\mathbb{R}^{d \times d}$ space as $\mathbf{J_d}$. The group of orthonormal transformations for a $d$ dimensional vector is given as $\mathcal{O}(d) = \{\mathbf{Q} \in \mathbb{R}^{d \times d} : \mathbf{Q^T Q} = \mathbf{QQ^T} = \mathbf{I_d}\}$. Furthermore, we denote the set of right orthonormal matrices as $S(m, n) = \{\mathbf{S} \in \mathbb{R}^{m \times n} : \mathbf{SS^T} = \mathbf{I_m}\}$

### 3.1 FEATURE GRAM MATRIX DISTANCE

We define the Feature Gram Matrix Distance (FG) as the Frobenius norm of the difference between the Gram matrices of the teacher and student features

$$\mathcal{D}_{FG}(\mathbf{R_t}, \mathbf{R_s}) = \left\| \mathbf{R_t R_t}^T - \mathbf{R_s R_s}^T \right\|_F = \|\mathbf{K_t} - \mathbf{K_s}\|_F \tag{1}$$

It is easy to see that $\forall i, j \langle \mathbf{w_i}, \mathbf{w_j} \rangle = \langle \mathbf{v_i}, \mathbf{v_j} \rangle$ if and only if $\mathcal{D}_{FG}(\mathbf{R_t}, \mathbf{R_s}) = 0$.

For our theoretical analysis, we assume that a student model is perfectly geometrically aligned with the teacher model if $\mathcal{D}_{FG} = 0$. We note perfect alignment, implies that $\mathbf{K_t} = \mathbf{K_s}$, as such their ranks must also be equal. Hence, while $d_t > d_s$, we are implicitly assuming that feature directions live almost exclusively in a low-rank subspace that is at most $d_s$ dimensions.

### 3.2 LEARNED PROJECTION BASED DISTANCE

A common way to avoid the pitfall of dimension mismatch between teacher and student models is to learn a linear projection matrix, which has been extensively employed in previous works. (Jiao et al., 2020; Chen et al., 2022; Miles et al., 2024). Formally, the learned projection distance is defined as

$$\mathcal{D}_{LinProj}(\mathbf{R_t}, \mathbf{R_s}) = \min_{\mathbf{P} \in \mathbb{R}^{d_t \times d_s}} \|\mathbf{R_s P} - \mathbf{R_t}\|_F \tag{2}$$

### 3.3 CENTERED KERNEL ALIGNMENT (CKA)

Centered Kernel Alignment was initially proposed in Kornblith et al. (2019) to compute a metric for the similarity between neural networks, and has subsequently been employed for distillation in image (Saha et al., 2022; Zhou et al., 2024) and language models (Jung et al., 2023; Dasgupta & Cohn, 2025). The construction of CKA allows for the use of any positive definite kernels and includes a rich mathematical construction through Reproducing Kernel Hilbert Spaces. However, in practice, CKA is almost always constructed using a simple linear kernel. Therefore, we will also be using linear CKA in this work. Formally, CKA is defined as

$$CKA(\mathbf{R_t}, \mathbf{R_s}) = \frac{\text{tr}(\mathbf{K_t K_s})}{\sqrt{\text{tr}(\mathbf{K_t K_t})}\sqrt{\text{tr}(\mathbf{K_s K_s})}}$$

CKA lies between 0 and 1, with the CKA of 1 implying perfect alignment. For consistency with our other distance based measures, we define a distance based on CKA as:

$$\mathcal{D}_{CKA}(\mathbf{R_t}, \mathbf{R_s}) = 1 - CKA(\mathbf{R_t}, \mathbf{R_s}) \tag{3}$$

## 3.4 PROCRUSTES DISTANCE

The Procrustes distance (Schönemann, 1966) is a key measure in statistical shape analysis (Kendall, 1977), where the focus lies on comparing the geometry of point sets in any particular space. Procrustes distance has been recently introduced as a suitable measure for comparing neural networks based on their representations. (Williams et al., 2021; Duong et al., 2023), however it remains unused in a distillation setting. Formally, it is defined as

$$\mathcal{P}(\mathbf{R_t}, \mathbf{R_s}) = \min_{\mathbf{Q} \in O(d_s)} \|\mathbf{R_s}\mathbf{Q} - \mathbf{R_t}\| \tag{4}$$

This definition is not well-defined in the above form when $d_s \neq d_t$. However, a kernel-based reformulation of the Procrustes distance exists that is exactly equivalent to the original formulation when the dimensions are equal, and serves as a natural generalization when the dimensions differ (Harvey et al., 2024a). So, we use this formulation.

$$\mathcal{D}_{\mathcal{P}}^2(\mathbf{R_t}, \mathbf{R_s}) = \operatorname{tr}(\mathbf{K_t}) + \operatorname{tr}(\mathbf{K_s}) - 2\left\|\mathbf{R_s}^T\mathbf{R_T}\right\|_* \tag{5}$$

The proof for this equivalence is omitted for brevity. We point the interested reader to Theorem 1 of Harvey et al. (2024a) for the full proof of equivalence.

## 4 THEORETICAL RESULTS

**Theorem 1.** *Let $\mathbf{R_t}$ and $\mathbf{R_s}$ be centered, unit norm matrix of feature activations, such that $\mathcal{D}_{FG} = 0$ and $\mathcal{D}_{CKA} = 0$. For any $\epsilon \in [0, 1]$, we can construct another set of points $\tilde{\mathbf{R}}_{\mathbf{t}}$ such that $\mathcal{D}_{CKA}(\tilde{\mathbf{R}}_{\mathbf{t}}, \mathbf{R_s}) \leq \epsilon$, but $\mathcal{D}_{FG}(\tilde{\mathbf{R}}_{\mathbf{t}}, \mathbf{R_s}) = \sqrt{\epsilon}\left\|\mathbf{R_t}\mathbf{R_t^T} - \mathbf{J_n}\right\|_F$*

*Proof.* We provide a proof sketch here and delegate the full proof to the appendix.

Let $\tilde{\mathbf{K}}_{\mathbf{t}} = (1 - \epsilon)\mathbf{K_t} + \epsilon\mathbf{J_n}$. Note that, as a sum of positive semi-definite matrices, $\tilde{\mathbf{K}}_{\mathbf{t}}$ is a positive semi-definite matrix, as such there must be a set of points within the unit sphere in of dimension $d_t$ that construct this Gram matrix. We denote these sets of points as $\tilde{\mathbf{R}}_{\mathbf{t}}$

By some algebra, we can see that $\mathcal{D}_{CKA}(\tilde{\mathbf{R}}_{\mathbf{t}}, \mathbf{R_s}) \leq \epsilon$, however $\mathcal{D}_{FG} = \sqrt{\epsilon}\left\|\mathbf{R_t}\mathbf{R_t^T} - \mathbf{J_n}\right\|_F$  □

*Remark.* While $\sqrt{\epsilon}$ might seems like a reasonably close bound, note that $\left\|\mathbf{R_t}\mathbf{R_t^T} - \mathbf{J_n}\right\|_F$ can be in $O(n)$ based on the feature geometry of $\mathbf{R_t}$. In the case of over parameterized models, the $n >> d$ is an implicit assumption, i.e the number of features can eclipse the dimensionality of representations. In particular, as shown in Figure 1 if $\mathbf{R_t}$ consists of two canonical directions that are opposite of each other, CKA can incorrectly imply equivalence in alignment with a higher order feature structure.

**Theorem 2.** *Let $\mathbf{R_t}$ and $\mathbf{R_s}$ be centered, unit norm matrix of feature activations. $\mathcal{D}_{LinProj} = 0 \Rightarrow \mathcal{D}_{FG} = 0$ if and only if the optimal linear projector is in the set of right Orthogonal Matrices, i.e $\mathbf{P} \in S(d_s, d_t)$*

*Proof.* First, let $\mathbf{P} \in S(d_s, d_t)$ so that $\mathbf{PP^T} = \mathbf{I_{d_s}}$. Now, it is easy to see that $\mathcal{D}_{LinProject} = 0$ implies that $\mathbf{R_s}\mathbf{P} = \mathbf{R_t}$. Now, we can see

$$\mathcal{D}_{FG} = \left\|\mathbf{R_s}\mathbf{R_s^T} - \mathbf{R_s}\mathbf{PP^T}\mathbf{R_s}\right\|_F = \left\|\mathbf{R_s}\mathbf{R_s^T} - \mathbf{R_s}\mathbf{R_s^T}\right\|_F = 0$$

Now, assume that $\mathbf{P} \notin S(d_s, d_t)$. $\mathcal{D}_{FG} = 0$ only if $\mathbf{R_s}\mathbf{PP^T}\mathbf{R_s} = \mathbf{R_s}\mathbf{R_s^T}$. This implies $\mathbf{R_s}(\mathbf{I_s} - \mathbf{PP^T})\mathbf{R_s^T} = 0$. For non-trivial values of $\mathbf{R_s}$ and if $\mathbf{P} \notin S(d_s, d_t)$, this means that

$\mathbf{R_s}(\mathbf{I_{d_s}} - \mathbf{PP^T}) = 0$, i.e $(\mathbf{I_{d_s}} - \mathbf{PP^T})$ must be entirely contained in the null-space of $\mathbf{R_s}$. When $\mathbf{R_s}$ is full rank, this implies that the $\mathbf{PP^T} = I_{d_s}$ which implies that $\mathbf{P} \in S(d_s, d_t)$. $\qquad\square$

*Remark.* The above theorem can be relaxed slightly if we can make further assumptions about $\mathbf{R_s}$. In particular, if the row-space of $\mathbf{R_s}$ is contained within the eigenspace of $\mathbf{PP^T}$ with the corresponding value of 1, we can say that even with $\mathbf{P} \notin S(d_s, d_t)$, $\mathcal{D}_{LinProj} = 0 \Rightarrow \mathcal{D}_{FG} = 0$. In general, this is a strong assumption to make and any spectral restrictions on the projection matrix is not common practice. So, we have included the proof and analysis for this scenario in the appendix.

Intuitively Theorem 2 tells us that restricting the possible output space of the learned linear projection to right orthogonal matrices, or increasing the eigenspace of corresponding to the eignevalue of 1 is a necessity in ensuring the optimal correlation between the feature structure and the projection based loss. Note that if $\mathbf{P}$ is restricted to be right orthogonal, the Projection based loss in Equation 2 bares significant similarity to the Procrustes distance in Equation 4.

**Theorem 3.** *Let $\mathbf{R_t}$ and $\mathbf{R_s}$ be centered, unit norm matrix of feature activations. $\mathcal{D}_{\mathcal{P}} = 0 \Leftrightarrow \mathcal{D}_{FG} = 0$*

*Proof.* We sketch the proof and defer details to the appendix. For the forward direction, we use the definition of nuclear norm to decompose $\left\|\mathbf{R_s^T R_t}\right\|_*$ as $\mathrm{tr}(\mathbf{U^T R_s R_t V})$ where $\mathbf{U}$ and $\mathbf{V}$ come from the SVD of $\mathbf{R_s^T R_t}$. So, simplifying the definition of $\mathcal{D}_{\mathcal{P}}$ in 5, we get $\mathcal{D}_{\mathcal{P}} = \left\|\mathbf{R_s U} - \mathbf{R_t V}\right\|_F$.

We now need to prove that if $\mathbf{R_s U} = \mathbf{R_t V}$, then $\mathbf{R_t R_t^T} = \mathbf{R_s R_s^T}$, we use the properties of $\mathbf{U}$ and $\mathbf{V}$ as orthonormal matrices and argue that that the row space of $\mathbf{R_t}$ must be in the column space of $\mathbf{V}$, leading to the desired equality of Gram matrices.

In the reverse direction, we argue that if $\mathbf{R_s R_s^T} = \mathbf{R_t R_t^T}$, then the SVDs must match in singular values. So, $\left\|\mathbf{R_s^T R_t}\right\|_* = \left\|\mathbf{R_s R_s^T}\right\|_F = \left\|\mathbf{R_t R_t^T}\right\|$. $\qquad\square$

## 5 EXPERIMENTS

First, we empirically validate our theoretical claims that Procrustes is the better measure to optimize over in order to preserve feature structure. In particular, we consider the geometry where the teacher has features that are approximately orthogonal to each other.

To demonstrate the feasibility of our model in a realistic setting, we evaluate the effectiveness of geometry-aware feature distillation across model architectures, tasks, and training settings. We evaluate our method on both encoder-only (BERT) and decoder-only (OPT) architectures. These models are widely used in both research and production contexts, and have served as benchmarks for prior distillation efforts. (Sanh et al., 2020; Mukherjee & Hassan Awadallah, 2020; Sun et al., 2019; Gu et al., 2024; Dasgupta & Cohn, 2025)

### 5.1 SYNTHETIC EXPERIMENT

**Data setup:** To better mimic the dimensional mismatch in real models, we set the teacher dimension to be $d_t = 1000$ and $d_s = 500$. We randomly sample $n$ unit norm vectors, that are $\epsilon$-orthogonal to each other, i.e $\mathbf{v_1} \ldots \mathbf{v_n}$ such that $|\langle \mathbf{v_i}, \mathbf{v_j} \rangle| \leq \epsilon$ for all $i \neq j$. It is easy to construct these $n = 2^{\frac{\epsilon^2 d_t}{4}}$ such vectors by simply randomly sampling each coordinate in $\mathbf{v_i} \in \mathbb{R}^{1000}$ to be $+1$ or $-1$ with probability $1/2$, and normalizing them to unit length. We set $\epsilon = 0.2$, and thus get $n = 22,026$ vectors that are all $\epsilon-$ orthogonal to each other. These vectors become our teacher representations. Mathematical details on why this construction works is included in the appendix.

We project the teacher representations down to $d_s = 500$ dimension using a random projection matrix. We further evaluate a setting in which student representations are randomly generated, ensuring no correspondence with the teacher's features. We observe consistent results across both experimental setups, and include details in the appendix.

**Training process:** We perform gradient based optimization over the Gram matrix distance (Eq 1), Projection based distance (Eq 2), CKA (Eq 3) and Procrustes distance (Eq 5) where the gradients are computed only on the student representations. To make this minimization more realistic to the distillation setting, we perform the optimization over batches. We use a batch size of 256 and use

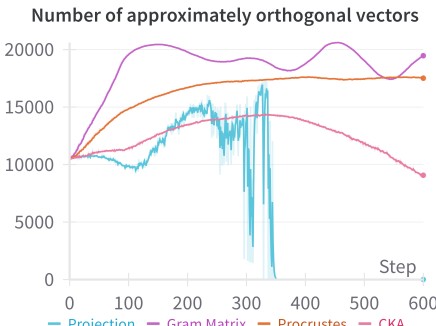

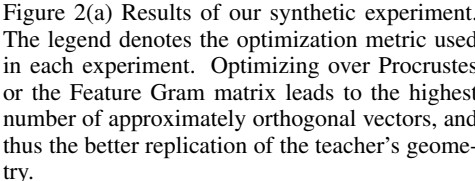

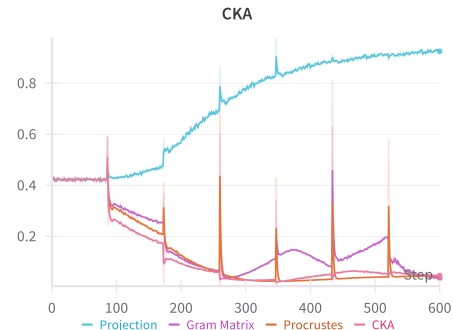

Figure 2(a) Results of our synthetic experiment. The legend denotes the optimization metric used in each experiment. Optimizing over Procrustes or the Feature Gram matrix leads to the highest number of approximately orthogonal vectors, and thus the better replication of the teacher's geometry.

Figure 2(b) The value of CKA while optimizing over the metrics. CKA converges close to 0 when optimized over all metrics, with the exception of linear projection.

ADAM (Kingma & Ba, 2015) with a learning rate of 0.01. Our training is performed for 7 epochs over five randomized seeds.

**Evaluation metrics:** We evaluate our optimization based on the number of $\epsilon$- orthogonal vectors in the student representations during a point in the optimization process. We formulate the problem of computing the maximum number of approximately orthogonal vector as a special case of the maximal independent set problem in graph theory. In particular, we consider the Gram matrix from student representations and consider an edge between two vectors if their inner product is more than $\epsilon$. While the maximal independent set problem is NP hard, we use Luby's algortihm (Luby, 1985), a classical randomized algorithm to compute an estimate for the size of the $\epsilon$-orthogonal vectors.

**Results:** As shown in Figure 2a, we find that Procrustes and the norm of the Gram Matrix leads to the highest number of approximately orthogonal vectors, implying that optimizing over them leads to the best replication of the teacher feature structure. We observe that the Procrustes method exhibits greater structural stability throughout the minimization process, whereas the Gram matrix demonstrates more pronounced fluctuations. We attribute the Gram-matrix fluctuations to batch noise, which can cause over-correction when inner products exceed $\epsilon$.

The inadequacies of CKA are quite apparent by this experiment. The number of orthogonal vectors goes down quite significantly even though the value of CKA is close to zero as seen in Figure 2b. Our findings corroborate claims that optimizing over Procrustes is in some sense "stronger" than optimizing over CKA (Cloos et al., 2024; Harvey et al., 2024b). Learned projection based distances demonstrates subpar performance; it is extremely noisy and underperforms even after optimization.

## 5.2 ENCODER-ONLY MODEL FOR CLASSIFICATION

**Dataset & Tasks:** We experiment on the GLUE benchmark (Wang et al., 2018). Specifically, we use three tasks within GLUE: CoLA (Warstadt et al., 2019), MRPC (Dolan & Brockett, 2005) and RTE (Bentivogli et al., 2009). CoLA involves predicting whether a sequence of words is a grammatical English sentence, and is evaluated using Matthews correlation coefficient (MCC) (Matthews, 1975). MRPC contains two sentences and the task involves predicting if they are semantically equivalent. Since the dataset is imbalanced, we report both accuracy and F1 score. RTE involves an entailment challenge; given a premise sentence and a hypothesis sentence, the task is to predict whether the premise entails the hypothesis. We evaluate RTE using classification accuracy. These tasks were chosen from the 9 GLUE benchmark tasks because they had the greatest discrepancy in performance between teacher and student model after five epochs of fine-tuning.

**Loss function:** Our loss function takes the form of

$$\mathcal{L} = \gamma \mathcal{L}_{CE}\left(f_S, \hat{y}\right) + \alpha \mathcal{L}_{\text{sim}}\left(\phi_T(f_T), \phi_S(f_S)\right) + (1-\alpha)\mathcal{L}_{\text{KD}}\left(f_S, f_T\right) \tag{6}$$

$\mathcal{L}_{CE}$ represents the cross entropy loss of the student logits with respect to output labels, $\mathcal{L}_{\text{sim}}$ is either $\mathcal{D}_{CKA}$ or $\mathcal{D}_{\mathcal{P}}$ and $\mathcal{L}_{KD}$ is the KL divergence between student and teacher logits.

$\gamma \in \{0, 1\}$ indicates whether we are including supervised cross entropy loss, and $\alpha \in [0, 1]$ controls the interplay between hidden layer and last layer similarities. $f_S$ and $f_T$ are outputs, including hidden representations, of student and teacher models. $\phi$ is a function that extracts hidden layers from the model. For ease of notation, if $\phi_T = (a, b)$, it is extracting hidden representations from the $a^{\text{th}}$ and $b^{\text{th}}$ layers of the model.

**Model details:** We perform all our distillation tasks on the BERT model. (Devlin et al., 2019). As in common in most distillation studies, we use pre-trained BERT-large model, which has 24 encoder layers, as the teacher model and pre-trained BERT-base model with half the layers removed as as the student model. We fine-tune the pre-trained BERT-large model for 5 epochs on each task, and use this fine-tuned model as the teacher for distillation. The student is not fine-tuned on any tasks.

**Training details:** We use a context size of 128, which aligns with most samples from the datasets. We optimize using ADAM (Kingma & Ba, 2015) with a learning rate of $2 \times 10^{-5}$ and a batch size per GPU of 64, with 2 NVIDIA A100 80 GB GPU. We use Hugging Face libraries (Wolf et al., 2020) to perform all our training and evaluation. We run an initial hyperparameter sweep over [0, 0.2, 0.4, 0.6, 0.8, 1] for the best value of $\alpha$ in Equation 6. Evaluations are reported after running distillation across the three tasks for 6 epochs. Furthermore, to ensure statistical significance in the performance of our distilled model, we use McNemar's test (McNemar, 1947; Dietterich, 1998) to compare all distilled models against the fine-tuned baseline. Unless otherwise noted, all results reported are statistically significant ($p < 0.05$)

**Multi Layer Distillation Results:** First, we present the results when distilling with Procrustes using all layers in network. Results are presented in Table 1. To ensure appropriate layers are matched, we match layer $n$ of the student model with layer $2n$ of the teacher model, as is common in previous works. We benchmark our results alongside Progressive KD (PKD) (Sun et al., 2019), DistillBERT (Sanh et al., 2020), MiniLMv2 (Wang et al., 2020), LinBERT and CKABERT. (Dasgupta & Cohn, 2025). All baseline results in Table 1 are taken from the original paper. Additionally, CKABERT and LINBERT from Dasgupta & Cohn (2025) is equivalent to $\mathcal{D}_{LinProj}$ and $\mathcal{D}_{CKA}$ Procrustes does better than all methods in CoLA while performing on par with MiniLLMv2 in MRPC, while MiniLLMv2 does better in RTE. Note that MiniLLMv2 involves aligning attention scores across all heads, and therefore is not perfectly comparable with other feature distillation methods.

| Method | COLA | RTE | MRPC |
|---|---|---|---|
| PKD (Sun et al.) | N/A | 65.9 | 86.2 |
| DistillBERT (Sanh et al.) | 51.3 | 59.9 | 87.5 |
| MinilLMv2 (Wang et al.) | 48.6 | **69.2** | **88.9** |
| LinBERT (Dasgupta et al.) | 46.5 | 61.0 | 87.0 |
| CKABert (Dasputa et al.) | 50.2 | 63.0 | 87.8 |
| Procrustes | **56.0** | 68.2 | **88.9** |

Table 1: Comparison of Procrustes distance as the distillation objective compared to other proposed feature distillation methods for BERT. Distillation is done over all layers.

**Procrustes does better, even with single layers:** For all tasks in this section, we assume $\phi_T = (12)$ and $\phi_S = (6)$, i.e we are aligning the middle layer of the teacher model with the middle layer of the student model. All results are noted after minimizing the loss function from Equation 6.

As shown in Table 2, including either Procrustes or CKA as $\mathcal{L}_{\text{sim}}$ alongside $\mathcal{L}_{KD}$ and $\mathcal{L}_{CE}$ increases the performance of the student model across all three tasks. Procrustes distance does better, with statistical significance, across all three tasks.

**Feature distillation, by itself, is disastrous:** When we remove KL divergence and fine-tuning loss entirely we see that the performance is significantly worse across all tasks and similarity functions. While leveraging the geometry of representations can steer the student model towards producing the correct output, it cannot by itself bias the model to produce the correct output. Some output information, either through teacher logits or supervised labels, are essential to ensure the model performs well on a particular task.

| Method | CoLA | RTE | MRPC |
|---|---|---|---|
| RD baseline | 0.00 | 0.50 | 68.0/80.9 |
| FT baseline | 51.02 | 61.73 | 81.6/87.7 |
| FT + KD | 51.52 | 63.89 † | 81.3/86.6 † |
| CKA only | 10.66 | 47.29 | 68.3/81.2 |
| CKA + KD | 52.03 | 64.62 † | 81.1/87.3 |
| CKA + KD + FT | 52.04 | 64.62 | 81.3/84.6 |
| Procrustes only | 11.94 | 56.31 | 68.3/81.2 |
| Procrustes + KD | 51.03 | 63.37 | 79.1/85.9 |
| Procrustes + KD + FT | **54.97** | **65.70** | **83.5/88.7** |

Table 2: Performance on MRPC, CoLA and RTE while distilling on a single layer of the GLUE dataset. **RD**: Random baseline, **FT**: Fine-tuning on labels, **KD**: Distillation on KL divergence of the last layer logits. † indicates results that are not statistically significant ($p \geq 0.05$).)

| Method | SelfInst | U-NI | S-NI |
|---|---|---|---|
| Seq-KD (Kim & Rush, 2016) | $10.81 \pm 0.001$ | $15.05 \pm 0.001$ | $7.33 \pm 0.0003$ |
| MiniLLM (Gu et al., 2024) | $10.83 \pm 0.002$ | $15.50 \pm 0.001$ | $7.34 \pm 0.0002$ |
| CKA ($\mathcal{D}_{\mathbf{CKA}}$) | $11.00 \pm 0.0002$ | $17.57 \pm 0.001$ | $8.30 \pm 0.0001$ |
| Gram matrix ($\mathcal{D}_{FG}$) | $11.07 \pm 0.0003$ | $\mathbf{17.60 \pm 0.001}$ | $8.31 \pm 0.0001$ |
| Procrustes ($\mathcal{D}_{\mathcal{P}}$) | $\mathbf{11.11 \pm 0.0003}$ | $17.59 \pm 0.001$ | $\mathbf{8.33 \pm 0.0001}$ |

Table 3: Rogue-L scores on instruction-following after distillation for 7000 steps on the Dolly dataset. Evaluations are reported with means and standard deviation with 5 random seeds. Seq-KD are MiniLLM are reported on models distilled using KL divergence by Gu et al. (2024)

## 5.3 Instruction following in LLMs

**Dataset & Task:** We experiment on the instruction following-task (Ouyang et al., 2022) and follow the same experimental setup as Gu et al. (2024). In particular, the model is prompted an instruction with a corresponding input, and is evaluated based on the correctness of the response.

Our teacher models are fine-tuned on the Databricks Dolly dataset (Conover et al., 2023), which consists of 15k instruction-response pairs. We measure the quality of the response using Rouge-L (Lin, 2004), which has been shown to be a good proxy for human-preference judgment in instruction-following tasks (Wang et al., 2022b).

We report our performance on three other instruction following datasets. These include: Self-Inst (Wang et al., 2022a), which consists of 252 instruction-following prompts,S-NI (Wang et al., 2022b), the test of SuperNaturalInstructions, which includes 9K samples across 119 tasks, and 10k randomly sampled instructions from U-NI, the UnaturalInstructions Dataset. (Honovich et al., 2022).

Our outputs are generated through multinational sampling with a temperature of 1 over five randomized seeds. We report the mean and standard deviation of the Rogue-L scores.

**Model details:** To be consistent with the baselines from Gu et al. (2024), we perform our experiments on the OPT model family (Zhang et al., 2022). The teacher model is a fine-tuned 40-layered OPT-13B on Dolly that we use from the MiniLLM Huggingface page [2], while the student model is a 24 layered OPT 1.3B.

**Training details:** Our loss function is the same as Equation 6, but with $\alpha = 1$, i.e we only optimize over the language modeling and feature distillation losses. We only align the last layers, primarily due to the increased computational complexity of aligning more layers. We use a context size of 1024 and optimize using a batch size of 4. We use 3 NVIDIA A100 80GB GPUs for training. We optimize using Adafactor (Shazeer & Stern, 2018), since it is more memory efficient and tends to have similar performance as the other optimizers in a low-batch setting (Marek et al., 2025). We report evaluations after performing distillation for 7000 steps.

---

[2] https://huggingface.co/MiniLLM/models

**Geometry preserving methods generally do better:** As seen in Table 3, we find that one Procrustes generally tends to get the best instruction-following performance in Self-Instruct and S-NI, while trailing close behind Gram Matrix difference in U-NI. In general, we find that both Procrustes and the Gram Matrix difference makes marginal, but statistically significant improvements compared to CKA, and make up to a $2\%$ improvement over previously proposed logit based distillation methods.

## 6 CONCLUSION

In this work, we take a critical look at prevailing distillation measures used in feature distillation, namely learned linear projection based distance and Centered Kernel Alignment. We present theoretical and empirical evidence demonstrating that these measures do not reliably preserve the feature geometry of teacher models. We introduce the Procrustes distance as a geometrically grounded loss function for feature distillation and show that it performs well in classification and instruction-following tasks.

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

# A THEORETICAL PROOFS

## A.1 PROOF OF THEOREM 1

We start by showing some elementary properties of positive semi-definite matrices that we will use in our proof

**Lemma 1.** *Let* $\mathbf{A}$ *and* $\mathbf{B}$ *be* $n \times n$ *positive semi-definite matrices. For any* $\alpha > 0$ *and* $\beta > 0$, $\alpha\mathbf{A} + \beta\mathbf{B}$ *is also positive semi-definite*

*Proof.* Since $\mathbf{A}$ and $\mathbf{B}$ are positive semi-definite, we have for all $\mathbf{x} \in \mathbb{R}^n$, $\mathbf{x}^{\mathbf{T}}\mathbf{A}\mathbf{x} \geq 0$ and $\mathbf{x}^{\mathbf{T}}\mathbf{B}\mathbf{x} \geq \mathbf{0}$

Let $\mathbf{C} = \alpha\mathbf{A} + \beta\mathbf{B}$. Now, for any $\mathbf{x} \in \mathbb{R}^n$, we have $\mathbf{x}^{\mathbf{T}}\mathbf{C}\mathbf{x} = \alpha\mathbf{x}^{\mathbf{T}}\mathbf{A}\mathbf{x} + \beta\mathbf{x}^{\mathbf{T}}\mathbf{B}\mathbf{x} \geq 0$ since $\alpha, \beta > 0$ and $\mathbf{A}$ and $\mathbf{B}$ are p.s.d.

$\square$

**Lemma 2.** *Let* $\mathbf{A}$ *and* $\mathbf{B}$ *be positive semi-definite matrices.* $\langle \mathbf{A}, \mathbf{B} \rangle = \mathrm{tr}(\mathbf{A}^T\mathbf{B}) \geq 0$

*Proof.* Since $\mathbf{A}$ is p.s.d we know that $\mathbf{A}^{\mathbf{T}} = \mathbf{A}$. So, now when $\mathbf{e_i}$ is the i-th basis vector,

$$\begin{aligned}
\langle \mathbf{A}, \mathbf{B} \rangle &= \mathrm{tr}(\mathbf{A}^{\mathbf{T}}\mathbf{B}) \\
&= \mathrm{tr}(\mathbf{A}\mathbf{B}) \\
&= \mathrm{tr}(\mathbf{A}^{1/2}\mathbf{A}^{1/2}\mathbf{B}) \\
&= \mathrm{tr}(\mathbf{A}^{1/2}\mathbf{B}\mathbf{A}^{1/2}) \\
&= \sum_{i=1}^{n} \mathbf{e_i}^{\mathbf{T}}(\mathbf{A}^{1/2}\mathbf{B}\mathbf{A}^{1/2})\mathbf{e_i} \\
&= \sum_{i=1}^{n} (\mathbf{A}^{1/2}\mathbf{e_i})^{\mathbf{T}}\mathbf{B}(\mathbf{A}^{1/2}\mathbf{e_i})
\end{aligned}$$

We use the cyclic property of the trace operator in the fourth step and the fact that $\mathbf{A}^{1/2}$ is also a symmetrical matrix in the fifth step. Now, since $\mathbf{B}$ is p.s.d and $\mathbf{A}^{1/2}\mathbf{e_i} \in \mathbb{R}^n$, each of terms in the summation is non-negative. Hence the sum is also non-negative. $\square$

We restate Theorem 1 below:

**Theorem 1.** *Let* $\mathbf{R_t}$ *and* $\mathbf{R_s}$ *be centered, unit norm matrix of feature activations, such that* $\mathcal{D}_{FG} = 0$ *and* $\mathcal{D}_{CKA} = 0$. *For any* $\epsilon \in [0, 1]$, *we can construct another set of points* $\tilde{\mathbf{R}}_{\mathbf{t}}$ *such that* $\mathcal{D}_{CKA}(\tilde{\mathbf{R}}_{\mathbf{t}}, \mathbf{R_s}) \leq \epsilon$, *but* $\mathcal{D}_{FG}(\tilde{\mathbf{R}}_{\mathbf{t}}, \mathbf{R_s}) = \sqrt{\epsilon} \left\| \mathbf{R_t}\mathbf{R_t^{T}} - \mathbf{J_n} \right\|_F$

We start with the assumption that there are $\mathbf{R_s}$ and $\mathbf{R_t}$ such that $\mathcal{D}_{CKA} = 0$ and $\mathcal{D}_{FG} = 0$. This implies that $\mathbf{K_s} = \mathbf{K_t}$.

Now, take $\epsilon \in [0, 1]$. We define $\tilde{\mathbf{K}}_{\mathbf{t}} = (1 - \epsilon)\mathbf{K_t} + \epsilon\mathbf{J_n}$ where $\mathbf{J_n}$ is the $n \times n$ all ones matrix. Note that $\tilde{\mathbf{K}}_{\mathbf{t}}$ is a valid Gram matrix in $d_t$ dimensions. To see this, note than both $\mathbf{K_t}$ and $\mathbf{J_n}$ are p.s.d matrices. Since both $\epsilon \in [0, 1]$, $1 - \epsilon \geq 0$. So, as conical combination of two positive semi definite matrices are positive semi definite, we know that $\tilde{\mathbf{K}}_{\mathbf{t}}$ is a positive semi-definite matrix. The rank of $\mathbf{K_t}$ is at most $d_s$ since it is equal to $\mathbf{K_s}$, which is a Gram matrix for vectors in $d_s$ dimension. The rank of $\mathbf{J_n}$ is 1. So, the subadditivity of matrix rank implies

$$\begin{aligned}
rank(\tilde{\mathbf{K}}_{\mathbf{t}}) &\leq rank(\mathbf{K_t}) + rank(\mathbf{J_n}) \\
&\leq d_s + 1 \leq d_t
\end{aligned}$$

The last equality follows since we explicitly require $d_t > d_s$, i.e the teacher feature dimension is greater than the student feature dimension.

Since $\tilde{\mathbf{K}}_\mathbf{t}$ is a psd matrix with $rank(\tilde{\mathbf{K}}_\mathbf{t}) \leq d_t$, there must be a set of $n$ points in $d_t$ whose Gram matrix is $\tilde{\mathbf{K}}_\mathbf{t}$. Furthermore, these vectors are all unit norm. To see this note that every diagonal entry in $\tilde{\mathbf{K}}_\mathbf{t}$ is 1. More explicitly, let $\tilde{k}_{i,i}$ be the $i$ th diagonal entry of $\tilde{\mathbf{K}}_\mathbf{t}$ and $k_{i,j}$ be the $i$th diagonal entry of $\mathbf{K}_\mathbf{t}$. Now, $\forall i \in [1 \ldots n]$ $\tilde{k}_{i,i} = (1-\epsilon)k_{i,i} + \epsilon = 1 - \epsilon + \epsilon = 1$, where we use the fact that $k_{i,i} = 1$, by the construction of $\mathbf{K}_\mathbf{t}$

Now, first, we show that $\left\|\tilde{\mathbf{K}}_\mathbf{t} - \mathbf{K}_\mathbf{s}\right\|_F = \sqrt{\epsilon}\,\|\mathbf{K}_\mathbf{t} - \mathbf{J}_n\|_F$. Note that since $\mathbf{K}_\mathbf{s} = \mathbf{K}_\mathbf{t}$,

$$
\begin{aligned}
\left\|\tilde{\mathbf{K}}_\mathbf{t} - \mathbf{K}_\mathbf{s}\right\|_F &= \|(1-\epsilon)\mathbf{K}_\mathbf{t} + \epsilon\mathbf{J_n} - \mathbf{K}_\mathbf{t}\|_F \\
&= \|\epsilon(\mathbf{K}_\mathbf{t} - \mathbf{J_n})\|_F = \sqrt{\epsilon}\,\|(\mathbf{K}_\mathbf{t} - \mathbf{J_n})\|_F
\end{aligned}
$$

On the other hand, let's show that $\mathcal{D}_{CKA} \leq \epsilon$. We'll begin by assuming the following identity holds and provide its proof later.

$$
\left\|\tilde{\mathbf{K}}_\mathbf{t}\right\|_F \geq (1-\epsilon)\,\|\mathbf{K}_\mathbf{t}\| - \epsilon n \tag{7}
$$

Now, for $\mathcal{D}_{CKA} \leq \epsilon$, we must have

$$
1 - \frac{\langle \mathbf{K}_\mathbf{t}, \tilde{\mathbf{K}}_\mathbf{t}\rangle}{\|\mathbf{K}_\mathbf{t}\|_F \left\|\tilde{\mathbf{K}}_\mathbf{t}\right\|_F} \leq \epsilon
$$

$$
\langle \mathbf{K}_\mathbf{t}, \tilde{\mathbf{K}}_\mathbf{t}\rangle \geq (1-\epsilon)\,\|\mathbf{K}_\mathbf{t}\|_F \left\|\tilde{\mathbf{K}}_\mathbf{t}\right\|_F
$$

$$
(1-\epsilon)\,\|\mathbf{K}_\mathbf{t}\|_F^2 + \epsilon\langle \mathbf{K}_\mathbf{t}, \mathbf{J_n}\rangle \geq (1-\epsilon)^2\,\|\mathbf{K}_\mathbf{t}\|_F^2 - \epsilon(1-\epsilon)n\,\|\mathbf{K}_\mathbf{t}\|_F
$$

$$
(1-\epsilon)\epsilon\,\|\mathbf{K}_\mathbf{t}\|_F^2 + \epsilon\langle \mathbf{K}_\mathbf{t}, \mathbf{J_n}\rangle \geq -\epsilon n(1-\epsilon)\,\|\mathbf{K}_\mathbf{t}\|_F
$$

This inequality is trivially true since the left hand contains entries that are non-negative, whereas the right hand is always negative.

Now, we prove the identity form 7. We start by noting that by the definition of $\tilde{\mathbf{K}}_\mathbf{t}$, $\epsilon\mathbf{J_n} = \tilde{\mathbf{K}}_\mathbf{t} - (1-\epsilon)\mathbf{K}_\mathbf{t}$. So, taking the squared Frobenius norm on both sides, we get,

$$
\begin{aligned}
\epsilon^2 n^2 &= \left\|\tilde{\mathbf{K}}_\mathbf{t} - (1-\epsilon)\mathbf{K}_\mathbf{t}\right\|_F^2 \\
&= \left\|\tilde{\mathbf{K}}_\mathbf{t}\right\|_F^2 - 2(1-\epsilon)\langle \tilde{\mathbf{K}}_\mathbf{t}, \mathbf{K}_\mathbf{t}\rangle + (1-\epsilon)^2\,\|K_t\|_F^2 \\
&\geq \left\|\tilde{\mathbf{K}}_\mathbf{t}\right\|_F^2 - 2(1-\epsilon)\left\|\tilde{\mathbf{K}}_\mathbf{t}\right\|_F \|\mathbf{K}_\mathbf{t}\|_F + (1-\epsilon)\mathbf{K}_\mathbf{t}_F^2 \\
&= \left(\left\|\tilde{K}_t\right\|_F - (1-\epsilon)\,\|K_t\|_F \left\|\tilde{K}_t\right\|_F - (1-\epsilon)\,\|K_t\|_F\right)^2 \\
\epsilon n &\geq \left|\left\|\tilde{K}_t\right\|_F - (1-\epsilon)\,\|K_t\|_F\right|
\end{aligned}
$$

Hence, $(1-\epsilon)\,\|K_t\|_F - \epsilon n \leq \left\|\tilde{K}_t\right\|_F$, thus completing the proof.

## A.2   Spectral assumption between $\mathbf{R_s}$ and $\mathbf{P}$ in Theorem 2

We expand on the discussion in the remarks of Theorem 2.

In proving the theorem, we showed that assuming $\mathcal{D}_{LinProj} = 0$, $\mathcal{D}_{FG} = 0$ only if $\mathbf{R_s}(\mathbf{I_{d_s}} - \mathbf{PP^T}) = 0$. This equation is true trivially if (i) $\mathbf{R_s} = 0$, (ii) $\mathbf{PP^T} = \mathbf{I_{d_s}}$, which means $\mathbf{P} \in S(d_s, d_t)$. We now elaborate a third condition, which is the most general. We must make an assumption that the row-space of $\mathbf{R_s}$ is contained entirely within the left eigenspace of $\mathbf{PP^T}$ with the eigenvalue of 1. More precisely:

**Lemma 3.** *For a matrix* $\mathbf{P} \in \mathbb{R}^{d_s \times d_t}$ *with a spectral decomposition,* $\mathbf{P} = \mathbf{U\Sigma V^T}$ *let* $\mathcal{U} = span(\mathbf{U}_{\sigma=\mathbf{i}})$ *be the space spanned by the columns of* $\mathbf{U}$ *with a corresponding singular value of 1.* $\mathbf{R_s}(I_{d_s} - PP^T = 0)$ *if and only if* $Row(\mathbf{R_s}) \subseteq \mathcal{U}$.

*Proof.* First, we start by showing that $Row(\mathbf{R_s}) \subseteq E_1 \Leftrightarrow \mathbf{R_s PP^T} = \mathbf{R_s}$ where $E_1 = span(\mathbf{E}_{\lambda=\mathbf{1}})$ is the left-eigenspace of $\mathbf{PP^T}$ with eigenvector $\lambda = 1$

For the forward direction, we use the definition of a left-eigenvector. Let $s^T$ be a row of $\mathbf{R_s}$. Since, this is in the rowspace of $\mathbf{R_s}$ and therefore, also in $E_1$, we have $s^T \mathbf{PP^T} = \lambda s^T = s^T$. So, $\mathbf{R_s PP^T} = \mathbf{R_s}$.

For the reverse direction, we note that for every row $s^T$ from $S$, since $\mathbf{R_s PP^T} = \mathbf{R_s}$, $s^T \mathbf{PP^T} = \lambda s^T$, i.e $s^T$ is in $E_1$. Since this holds for every row in $\mathbf{R_s}$, it must also hold for the row-space of $\mathbf{R_s}$. Hence $Row(\mathbf{R_s}) \subseteq E_1$

Now, we conclude by simply restating that the left-eigenspace of $E_1$ is the same space as $\mathcal{U}$. We do this by noting that $\mathbf{U}$ is the same as the matrix of eigenvectors of $\mathbf{PP^T}$. Since, $\mathbf{PP^T}$ is square symmetrical matrix, we have that the left-eigenvectors of $\mathbf{PP^T}$ are the transpose of it's right eigenvectors. In this case, since $\mathbf{U}$ is symmetric $\mathbf{U^T} = \mathbf{U}$. So, $\mathbf{U}_{\sigma=1} = \mathbf{E}_{\lambda=1}$, and consequently, $E_1 = \mathcal{U}$ □

This is a generalization of the theorem we present in the main paper. In particular if $\mathbf{P} \in S(d_s, d_t)$, then $\mathcal{U} = \mathbb{R}^{d_s}$. So, any set of vectors in $d_s$ dimension will be included in the left singular vector space of $\mathbf{P}$. This clarification implies that in the case that a structural condition is imposed on the student vectors (for instance, their row space spans a small subspsace), we can get an optimal linear projector that is not in $S(d_s, d_t)$. We view this as a largely unrealistic scenario in practical distillation settings, though it may offer value from a theoretical or analytical perspective.

A.3  PROOF OF THEOREM 3

We restate the theorem below:

**Theorem 3.** *Let* $\mathbf{R_t}$ *and* $\mathbf{R_s}$ *be centered, unit norm matrix of feature activations.* $\mathcal{D}_{\mathcal{P}} = 0 \Leftrightarrow \mathcal{D}_{FG} = 0$

*Proof.* We use a classical result relating the nuclear norm with singular value decomposition. The proof for this can be found in Horn & Johnson (1991) Theorem 3.4.1

$$\left\|\mathbf{R_s^T R_t}\right\|_* = \max_{\mathbf{U},\mathbf{V}} \operatorname{tr}\left(\mathbf{U R_s^T R_t V}\right)$$

where $\mathbf{U} \in \mathbb{R}^{d_s \times d_s}$ and $\mathbf{U^T U} = \mathbf{I_{d_s}}$ while $\mathbf{V} \in \mathbb{R}^{d_t \times d_s}$ and $\mathbf{V^T V} = \mathbf{I_{d_s}}$. We will use the fact the that $\mathbf{U}$ and $\mathbf{V}$ that maximize the above expressions are exactly the matrices from the singular value decomposition of $\mathbf{R_s^T R_t} = \mathbf{U\Sigma V}^T$

Plugging this into the definition of $\mathcal{D}_{\mathcal{P}}$, and using the fact that $\operatorname{tr}(\mathbf{XX^T}) = \|\mathbf{X}\|_F$, we get

$$\mathcal{D}_{\mathcal{P}} = \|\mathbf{R_s}\|_F + \|\mathbf{R_t}\|_F - 2\operatorname{tr}\left(\mathbf{U R_s^T R_t V}\right)$$
$$= \|\mathbf{R_s U} - \mathbf{R_t V}\|_F$$

First, we show that if $\mathcal{D}_{\mathcal{P}} = 0 \Rightarrow \mathcal{D}_{FG} = 0$. If $\mathcal{D}_{\mathcal{P}} = 0$, this means $\mathbf{R_s U} = \mathbf{R_t V}$, so that $\mathbf{R_s} = \mathbf{R_t V U^T}$. Now, $\mathbf{R_s R_s^T} = \mathbf{R_t V V^T R_t^T}$. Since $\mathbf{V}$ has orthonormal columns, $\mathbf{V V^T}$ is a projection matrix on the column space of $\mathbf{V}$. If we show that the row-space of $\mathbf{R_t}$ is a subspace of the column space of $\mathbf{V}$, we can say that $\mathbf{R_t V V^T R_t^T} = \mathbf{R_t R_t^T}$.

To see this we replace $\mathbf{R_s} = \mathbf{R_t V U}$ in the SVD of $\mathbf{R_s^T R_t}$, we get

$$\mathbf{U^T V^T R_t^T R_t = U\Sigma V^T}$$
$$\mathbf{V^T R_t^T R_t = \Sigma V^T}$$

So, the columns of $\mathbf{V}$ are the eigenvectors of $\mathbf{R_t^T R_t}$, which implies that the columns of $\mathbf{V}$ where the singular values in $\mathbf{\Sigma}$ are non-zero must be column space of $\mathbf{R_t^T}$, or the row-space of $\mathbf{R_t}$. Hence, the row space of $\mathbf{R_t}$ is contained within the column space of $\mathbf{V}$.

Finally, for the reverse direction, we assume $\mathcal{D}_{FG} = 0$, which implies $\text{tr}(\mathbf{R_s R_s^T}) = \text{tr}(\mathbf{R_t R_t^T})$. We need to show then that $\text{tr}(\mathbf{R_s R_s^T}) = \left\| \mathbf{R_s^T R_t} \right\|_*$ to conclude that $\mathcal{D}_{FG} = 0 \Rightarrow \mathcal{D}_{\mathcal{P}} = 0$

Let $\mathbf{R_s} = \mathbf{U_s \Sigma_s V_s^T}$ and $\mathbf{R_t} = \mathbf{U_t \Sigma_t V_t^T}$ be the SVD decompositions. Since $\mathbf{R_s R_s^T} = \mathbf{U_s \Sigma_s^2 U_s^T} = \mathbf{U_t \Sigma_t^2 U_t^T}$, $\mathbf{U}_s = \mathbf{U}_t$ and $\mathbf{\Sigma}_s = \mathbf{\Sigma}_t$. Hence $\text{tr}(\mathbf{R_s R_s^T}) = \text{tr}(\mathbf{\Sigma_s^2})$

Now, $\mathbf{R_s^T R_t} = \mathbf{V_s \Sigma_s U_s^T U_s \Sigma_s V_t^T} = \mathbf{V_s \Sigma_s^2 V_t^T}$. $\left\| \mathbf{R_s^T R_t} \right\|_*$ is just simply the sum of singular value, i.e $\text{tr}(\mathbf{\Sigma_s}^2)$ $\qquad\qquad\square$

# B  SYNTHETIC EXPERIMENT

## B.1  PROOF OF TEACHER VECTOR CONSTRUCTION

First, we prove that our construction of teacher vectors ensures that they are $\epsilon-$orthogonal to each other with high probability. More precisely:

**Lemma 4.** *Let $\mathbf{v_i} = [v_{i1}, v_{i2} \dots v_{in}] \in \mathbb{R}^n$ such that each $v_{ij} = 1/\sqrt{n}$ with probability $1/2$ and $-1/\sqrt{n}$ with probability $1/2$ independent of all other $v_{ij}$. For any $\epsilon > 0$ assume that we generate $k = 2^{c\epsilon^2 n}$ such vectors, with $c = \frac{1}{4\ln(2)}$ . Then we have $\Pr\left\{ \exists i, j \mid |\langle \mathbf{v_i}, \mathbf{v_j} \rangle| \geq \epsilon \right\} \leq \frac{1}{e^{\epsilon^2 n}}$*

*Proof.* We assume $i \neq j$ throughout the proof.

First we note that by the linearity of expectation and the independence of each term within the vectors,

$$\mathbb{E}\langle \mathbf{v_i}, \mathbf{v_j} \rangle = \sum_{k=1}^{n} \mathbb{E}(v_{ik} \cdot v_{jk}) = \sum_{i]1}^{n} \mathbb{E}(v_{ik}) \mathbb{E}(v_{jk}) = 0$$

$\qquad\qquad\square$

Now, use Hoeffding's concentration inequality Hoeffding (1963) to bound the inner product for a particular pair $i, j$. We use the fact that the inner product is a sum of random variables with zero expectation and each term is bounded below by $-\frac{1}{n}$ and above by $\frac{1}{n}$ to claim

$$\Pr\left\{ |\langle \mathbf{v_i}, \mathbf{v_j} \rangle| \geq \epsilon \right\} \leq 2\exp\left( -\frac{2\epsilon^2}{\sum_{i=1}^{n} 2/n^2} \right)$$
$$= 2\exp(-\epsilon^2 n)$$

Now, we use an union bound over all $\binom{k}{2} \leq \frac{k^2}{2}$ to claim that

$$\Pr\left\{ \exists i, j \mid |\langle \mathbf{v_i}, \mathbf{v_j} \rangle| \geq \epsilon \right\} \leq \frac{k^2}{2} 2\exp(-\epsilon^2 n)$$
$$= \exp(-\epsilon^2 n + 2c\epsilon^2 n \ln(2))$$
$$= \exp\left( \epsilon^2 n \left( \frac{\ln(2)}{2} - 1 \right) \right)$$
$$= \exp(-0.65\epsilon^2 n)$$

Hence, since the probability of any two vectors having an inner product greater than $\epsilon$ decays exponentially as $n$ grows, which means in high dimensions, our teacher vector constructions will be $\epsilon-$orthogonal to each other with high probability.

## B.2 Luby's algorithm

We use Luby's algorithm Luby (1985) as a simple and efficient proxy for the exact number of vectors that are $\epsilon$- orthogonal to each other. We use the Gram matrix and transform it into an adjacency matrix of a graph where two vectors share an edge if their inner product is greater than $\epsilon$. The maximal independent set problem from graph theory can now be applied to this problem to identify the maximum number of vectors such that none of them have inner product higher than $\epsilon$.

We give the pseudocode for Luby's algorithm below:

---

**Algorithm 1** Luby's Algorithm for Maximal Independent Set (MIS)

---

1: **procedure** LUBYMIS($G = (V, E)$)
2: $\quad I \leftarrow \emptyset$
3: $\quad V' \leftarrow V$
4: $\quad$ **while** $V' \neq \emptyset$ **do** $\qquad\qquad\qquad\qquad$ ▷ Step 1: Assign random priorities to active nodes
5: $\quad\quad$ **for all** $v \in V'$ **in parallel do**
6: $\quad\quad\quad$ Assign a random priority $p(v)$
7: $\quad\quad$ **end for**
8: $\quad\quad S \leftarrow \emptyset$ $\qquad\qquad\qquad\qquad$ ▷ Initialize a temporary set for newly selected nodes
9: $\quad\quad$ **for all** $v \in V'$ **in parallel do** ▷ Step 2: Select nodes with a higher priority than all their active neighbors
10: $\quad\quad\quad$ **if** $p(v) > p(u)$ for all neighbors $u \in N(v) \cap V'$ **then**
11: $\quad\quad\quad\quad S \leftarrow S \cup \{v\}$
12: $\quad\quad\quad$ **end if**
13: $\quad\quad$ **end for**
14: $\quad\quad I \leftarrow I \cup S$ $\qquad\qquad\qquad$ ▷ Step 3: Update the MIS and the set of remaining nodes
15: $\quad\quad V' \leftarrow V' \setminus (S \cup N(S))$ $\qquad\qquad\qquad$ ▷ Remove $S$ and its neighbors from $V'$
16: $\quad$ **end while**
17: $\quad$ **return** $I$
18: **end procedure**

---

## B.3 Loss Curves for Different Objectives

In the main text, we present the graphs with the dynamics of CKA and the number of approximate orthogonal vectors over the trainnig process. In this section we include graphs of all the measures we optimize through including Procrustes (Figure 3), Feature Gram (Figure 4) and learned linear projection (Figure 5)

Note that we do not compute the learned linear projection when optimizing with any other measure, as doing so without learning the linear projection itself would not be meaningful.

## B.4 Results when student vectors are randomly initialized

In the main text, we present the results when $\mathbf{R_s} = \mathbf{R_t}\mathbf{P}$ where $\mathbf{P}$ is a randomly initialized matrix. Instead, we could have initialized $\mathbf{R_s}$ as completely random unit norm vectors. In this section, we present the results for that case.

The number of approximate orthogonal vectors are shown in Figure 6. The dynamics of Procrustes loss (Figure 7, CKA loss (Figure 8) and feature Gram matrix loss (Figure 9) are also shown

Note that the key takeaways are still the same; CKA and learned linear projection are incapable of preserving the feature geometry, despite having low losses. We notice the same noisy, fluctuating number of approximately orthogonal vectors with the norm of the Gram matrix, while Procrustes is similarity stable. A key difference in this case is that even for Procrustes and Frobenius norm of the

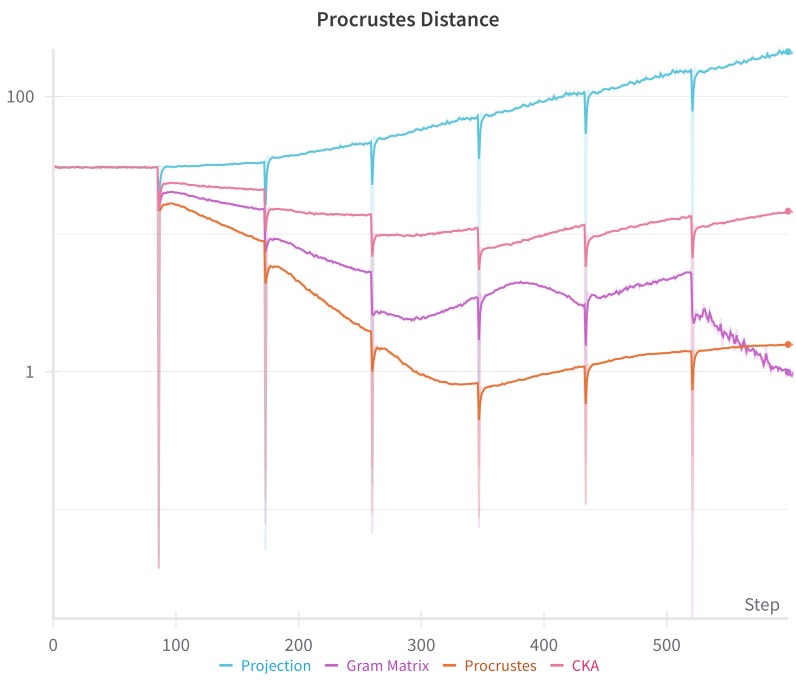

Figure 3: Dynamics of Procrustes distance throughout the synthetic training process when student vectors are initialized from a random projection

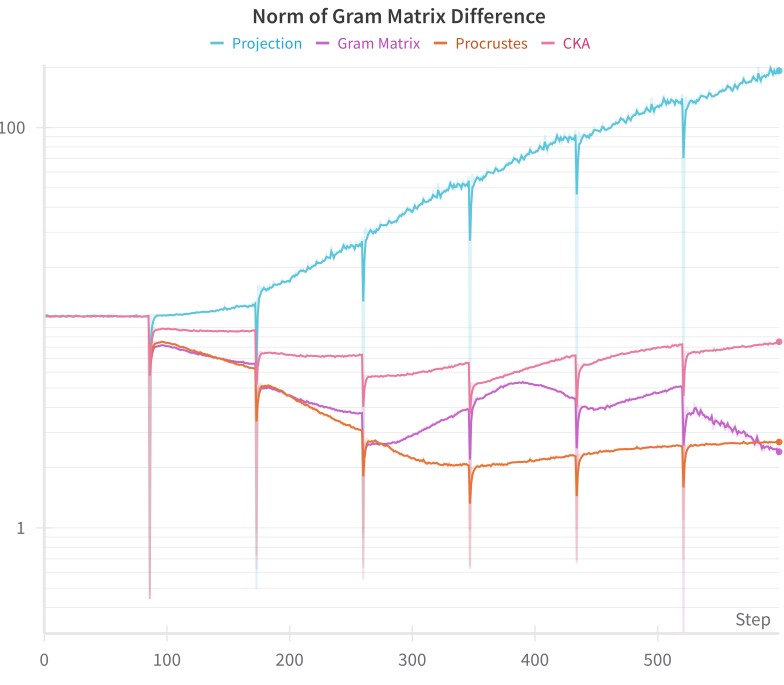

Figure 4: Dynamics of the norm of the difference in Feature Gram matrices throughout the synthetic training process when student vectors are initialized from a random projection

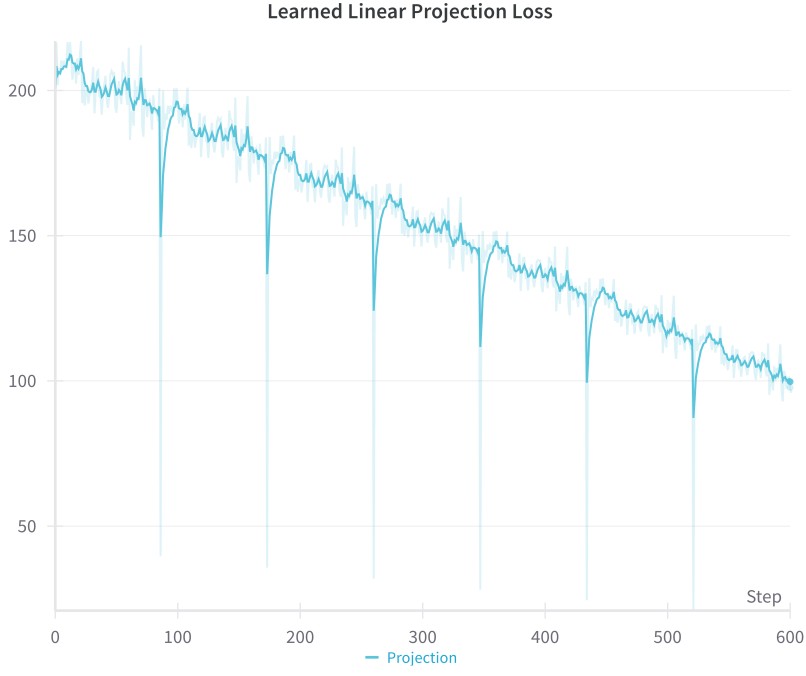

Figure 5: Dynamics of the learned linear projection loss throughout the synthetic training process when student vectors are initialized from a random projection.

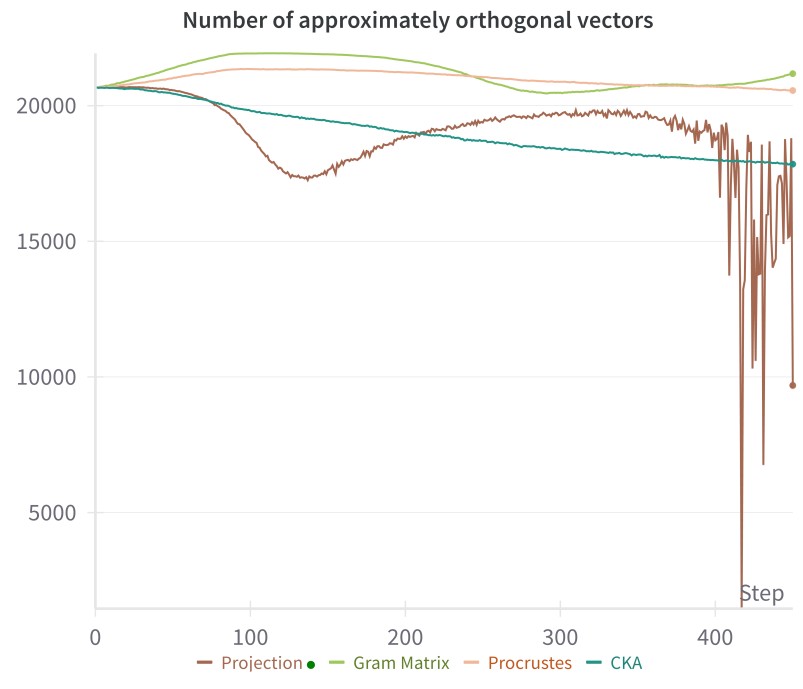

Figure 6: Dynamics of the number of approximate orthogonal vectors through the synthetic training process when the student vectors are randomly initialized

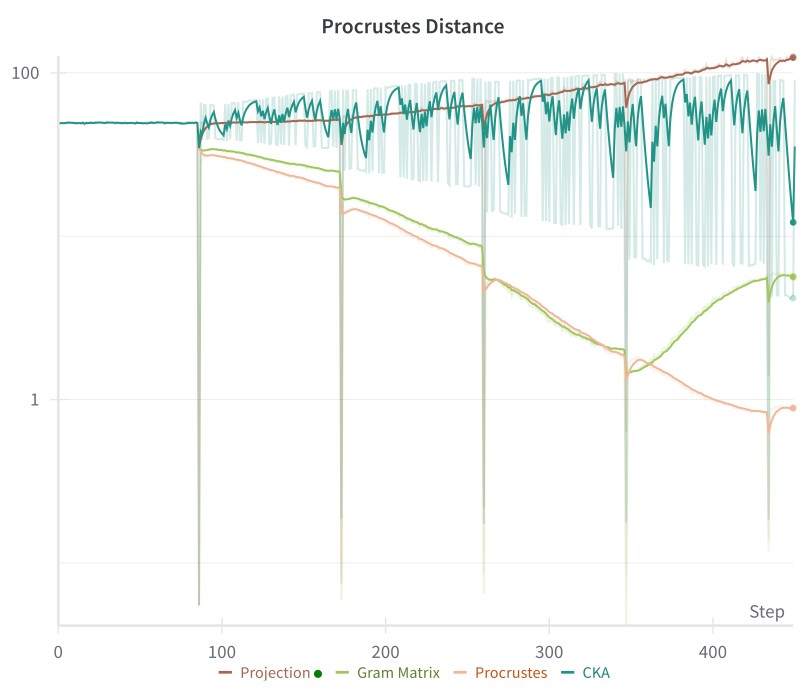

Figure 7: Dynamics of Procrustes distance through the synthetic training process when the student vectors are randomly initialized

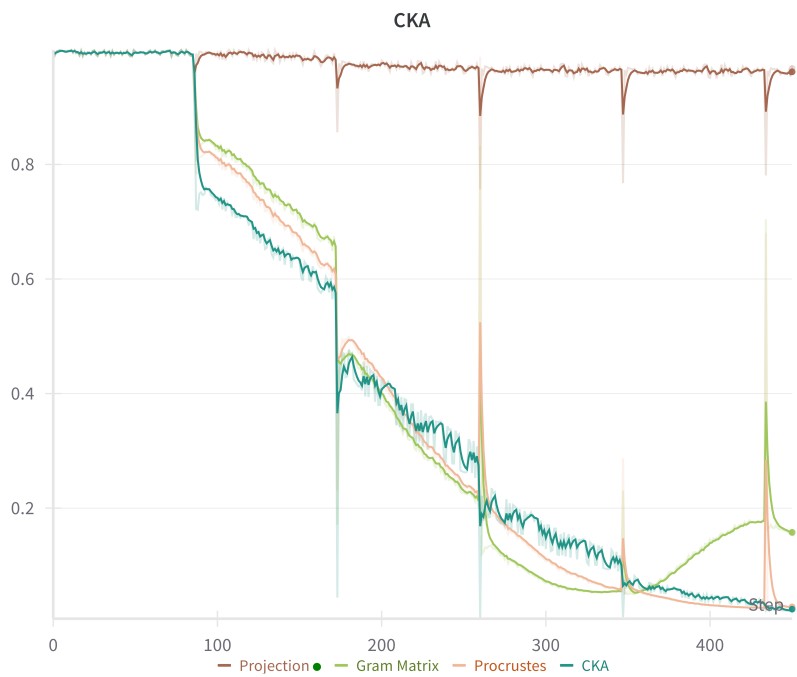

Figure 8: Dynamics of CKA through the synthetic training process when the student vectors are randomly initialized

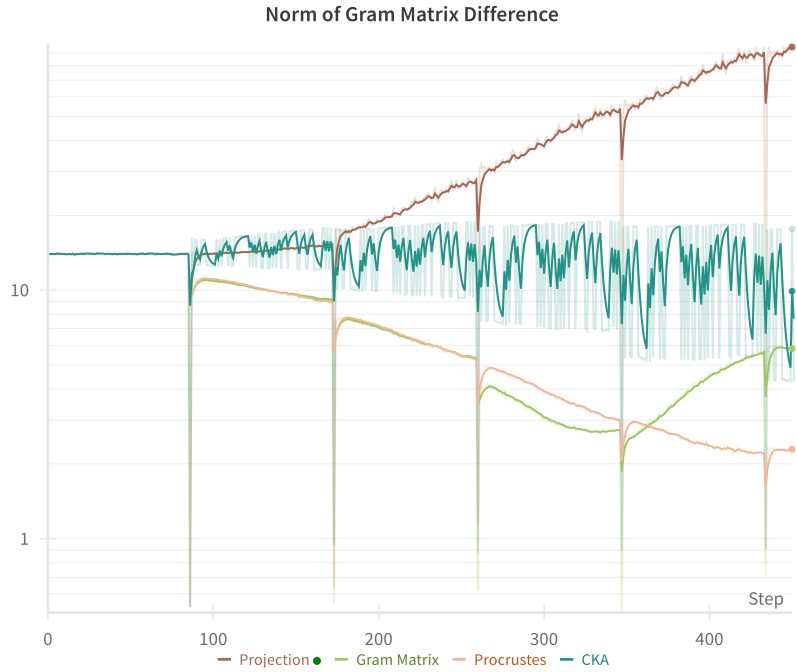

Figure 9: Dynamics of the norm of the difference in Feature Gram matrices through the synthetic training process when the student vectors are randomly initialized

Gram matrix, the performance seems to taper off as we keep optimizing. This is likely due to the fact that randomly generated student vectors are already nearly orthogonal at initialization, thereby limiting the extent to which the optimization process can further increase their orthogonality.

### B.5 Changing the dimensions of $d_s$ and $d_t$

In the main text, we use $d_t = 1000$ and $d_s = 500$. Here, we vary the dimensions and plot the number of $\epsilon$ orthogonal vectors. We first set $d_t = 1024$ and $d_s = 768$; the same dimensionality as the experiments in Section 5.2. The result are given in Figure 10. We find the dynamics largely similar to the one in Figure 2a and Figure 6 We also follow the same dimensionality as in Section 5.3, with $d_t = 5120$ and $d_s = 2560$, and show the results in Figure 11. In this setup, we find the optimization process is less smooth; the number of approximately orthogonal vectors when optimized with Procrustes and Gram matrix tends to fluctuate rather than exhibit the stable growth shown before. However, the ability for Procrustes and Gram matrix to better represent the orthogonal geometry is still present; both CKA and Linear Project tend to collapse very quickly.

## C Instruction following task

### C.1 Prompt template

During both training and evaluation, we use the following prompt wrapper to ensure uniformity across the various datasets.

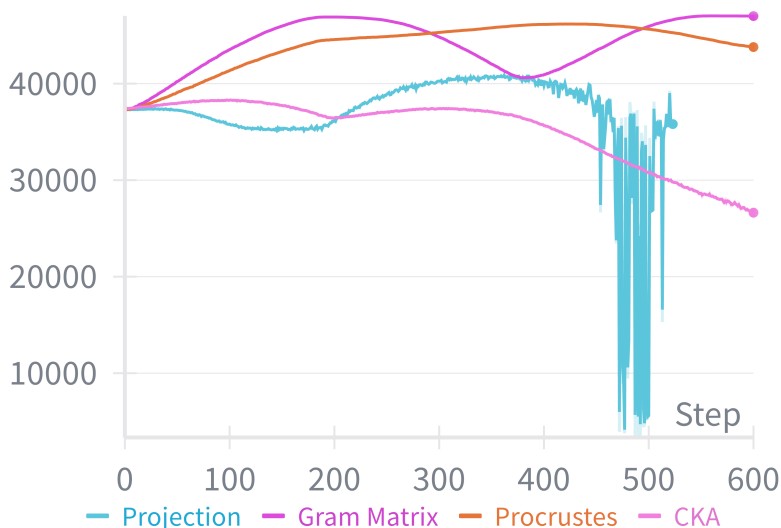

Figure 10: Dynamics of the number of orthogonal vectors through synthetic training when $d_t = 1024$ and $d_s = 768$.

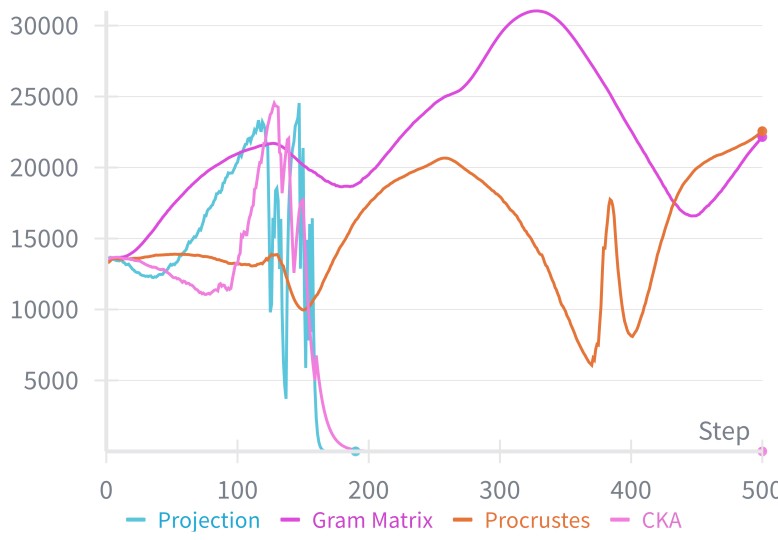

Figure 11: Dynamics of the number of orthogonal vectors through synthetic training when $d_t = 5120$ and $d_s = 2560$.

Below is an instruction that describes a task.
Write a response that appropriately completes the request.

### Instruction:
{instruction}

### Input:
{input}

### Response:

Figure 12: The prompt wrapper for training and evaluation.

