# OpenReview forum: "Knowledge distillation through geometry-aware representational alignment"
_ICLR.cc/2026/Conference — Submitted to ICLR 2026_

### Official Review · Reviewer_p4dj · 2025-10-30

**Soundness:** 1
**Presentation:** 2
**Contribution:** 2
**Rating:** 2
**Confidence:** 4

**Summary:**

This paper introduces two new forms of knowledge distillation that, informed by the linear representation hypothesis, minimise either the procrustes distance or the Frobenius norm of the feature gram matrix to improve distillation by improving representation alignment between students and teachers. The paper provides theoretical results, synthetic experiments and larger-scale experiments to show that this form of distillation shows statistically significant but marginal performance gains across a range of evaluations, namely: COLA, RTE, MRPC, SelfInst, U-NI and S-NI. Their core contribution is that minimising loss to preserve geometric properties of the teacher model's representation is a more effect way to perform knowledge distillation.

**Strengths:**

1. I appreciate the effort that has gone into exploring the geometry alignment that is enabled between students and teachers, I think that it is an interesting angle and the relation to the Linear Representation Hypothesis (LRH) is a good motivation for this work.
2. The use of statistical significance is a positive as many works do not conduct this type of analysis.
3. Comparison to lots of over methods is a benefit of the paper, however the lack of comparison to $D_{LinProj}$ is missed which feels like an important counterfactual to the LRH argument presented in this paper.
4. It is appreciated two language models families (BERT and OPT) are evaluated in this paper showing the benefits of procrustes distance for both classification and instruction following tasks.

**Weaknesses:**

While the finding that minimising either the procrustes distance or the Gram matrix can improve performance is interesting, it is unclear whether the linear representation hypothesis is the main reason why this occurs. The paper has some issues with experimental consistency and data formatting, which make it difficult to track the findings. Also, the missing evaluation of hyperparameter sweeps over the $\alpha$ term in the loss reduces certainty over the findings presented in the paper.


1. **Lack of hyperparameter sweeping on Alpha ($\alpha$)**: In the loss function introduced in equation (6) there is a balancing between knowledge distillation and the distance metric ( Procrustes ($D_{P}$), Gram Matrix ($D_{FG}$) or CKA ($D_{CKA}$)) used, it is unclear what the relationship is between the $\alpha$ hyperparameter and performance. I would like to see an experiment that varies this $\alpha$ hyperparameter keeping it constant across controls to show how $\alpha$ explicitly impacts performance of $D_{P}$, $D_{FG}$ or $D_{CKA}$, if alignment of orthogonality of the student and teacher is important, it should be expected that increasing the $\alpha$ value would put more preference to optimising towards this goal and improve performance. Also including results for $D_{LinProj}$ that show it is ineffective in boosting performance could improve the strength of the LRH argument.

2. **Section 5.1 Synthetic Experiment**: In sub figure A, it can be observed that the number of ε- orthogonal vectors in the student model is consistently the highest for $D_{FG}$ over $D_{P}$, however, the results in Table 3 suggest that $D_{P}$ persistently outperforms. As a result, it seems that purely aligning orthogonality between the student and the teacher does not fully explain the benefits of using $D_{P}$.

3. **Missing result in Table 1 for PKD**: There appears to be a missing result for the method PKD on the COLA evaluation - it is unclear why this result is missing, and I cannot find an explanation for its omission in the paper.

4. **Lack of standard deviations in Tables 1 and 2**: There appears to be no standard deviations presented in two of the main tables in the paper. While Table 3 does have standard deviations recorded, it is unclear why this is omitted for Tables 1 and 2.

5. **Lack of comparison with $D_{P}$**: In Table 2, it is unclear why we only compare to the procrustes distance ($D_{P}$) to  ($D_{CKA}$) when the Gram matrix ($D_{FG}$) is also proposed as an effective geometry preserving metric by this paper.

6. **Minor spelling mistake**: See line 345 'Our finds corroborate' which should be  'Our findings corroborate'  and line 346 'extremely noise' which should be  'extremely noisy'.

**Questions:**

1. How does Alpha ($\alpha$) impact the knowledge transfer and corresponding performance?

2. In Table 2, can you provide results for Procrustes + FT and CKA + FT? It is important to establish if minimising cross-entropy loss with these similarity metrics alone can improve performance; otherwise, it could be the case that these metrics merely add further regularisation to KD without directly improving performance through better or worse geometry alignment.

3. Can you align your findings with the recent distillation scaling laws [1] to show how student capacity impacts the geometry alignment?

4. Is there a specific relationship that emerges between which layers are aligned and how this impacts performance? Can ablations show that the layer presented represents a consistent trend for transfer?

5. In the appendix (Line 1188) there is a statement *'CKA and learned linear projection are incapable of preserving the feature geometry'* - given this and that in Table 2 we observe that CKA can correspond with statistically significant improvements in accuracy (see CKA+KD+FT) does this not show that preserving feature geometry is not a necessary part of performance increase and that performance benefits may not be tied to the LRH generally as this work posits?

References:

[1] Busbridge, D., Shidani, A., Weers, F., Ramapuram, J., Littwin, E. and Webb, R., 2025. Distillation scaling laws. arXiv preprint arXiv:2502.08606.

---

> ### Author Response · Authors · 2025-11-20
> **response (2/2)**
>
> **On the causal link between feature geometry/LRH and distillation performance**
>
> >Section 5.1 Synthetic Experiment: In sub figure A, it can be observed that the number of ε- orthogonal vectors in the student model is consistently the highest for $D_{FG}$ over
> $D_P$, however, the results in Table 3 suggest that
> $D_{P}$ persistently outperforms. As a result, it seems that purely aligning orthogonality between the student and the teacher does not fully explain the benefits of using $D_{FG}$.
>
> > In the appendix (Line 1188) there is a statement 'CKA and learned linear projection are incapable of preserving the feature geometry' - given this and that in Table 2 we observe that CKA can correspond with statistically significant improvements in accuracy (see CKA+KD+FT) does this not show that preserving feature geometry is not a necessary part of performance increase and that performance benefits may not be tied to the LRH generally as this work posits?
>
>
> We want to clarify that it is not our intent to argue that LRH is strictly necessary in improving the distillation performance. We use the LRH as a motivating example of how feature geometry has been shown to embed semantic information, which we argue can then be used for better downstream performance. We do not claim that LRH or feature geometry will, by itself, lead to an increase in performance. In fact, as we show in 5.2, just aligning feature geometry (using either CKA or Procrustes), without also using cross entropy and KL divergence leads to a worse performance across all tasks. However, do we find that when Procrustes is aligned with fine-tuning and KL divergence, it tends to maximize the performance across both encoder-only and decoder-only model architectures.
>
> We apologize for the generality in the statement the reviewer has included. We meant that CKA and learned linear projections are incapable of preserving the approximate orthogonal geometry of the synthetic experiments. We will rephrase it to make this more explicit.
>
> **Clarifying the baseline results**
> >Missing result in Table 1 for PKD: There appears to be a missing result for the method PKD on the COLA evaluation - it is unclear why this result is missing, and I cannot find an explanation for its omission in the paper.
>
> We apologize for this oversight in the paper. The scores for PKD (and all other baselines we include in section 5.2) come directly from the original paper. The authors of [1] do not include CoLA scores in their paper. We have clarified this in section 5.2 (Line 407)
>
> We also note that LINBERT from [2] is equivalent to our definition of $D_{LinProj}$. We further clarify this in section 5.2 (Line 408)
>
>
> **Statistical significance presented in the results**
>
> >Lack of standard deviations in Tables 1 and 2: There appears to be no standard deviations presented in two of the main tables in the paper. While Table 3 does have standard deviations recorded, it is unclear why this is omitted for Tables 1 and 2.
>
> We clarify that the standard deviations reported in Table 3 correspond to generation variation across five randomized seeds since we evaluate free form generation and adherence to instruction following in this task. Tables 1 and 2 correspond to classification tasks, and thus they do not exhibit the same kind of variance across generations. We nevertheless quantify the reliability of improvements in Table 2, by performing the McNemar's test on the truth tables output by the classification task. This is done to show that the models are significantly different from standard fine-tuned models.
>
> **Distillation scaling**
> > Can you align your findings with the recent distillation scaling laws [1] to show how student capacity impacts the geometry alignment?
>
> We thank the reviewer for pointing out the interesting connection to scaling laws. Given our current compute limits, the experiments we perform are primarily based on one teacher-student pair size, and computing a full scaling curve for distilling feature geometry is beyond our computational scope.
>
>
> We sincerely thank the reviewer for their careful reading and constructive feedback. We hope that these clarifications and additions help better highlight our main contributions, and we respectfully hope the updated version may positively inform your final assessment.

---

> ### Author Response · Authors · 2025-11-20
> **response (1/2)**
>
> We thank the reviewer for their detailed and constructive feedback and for the close reading of our paper. We have included our response to their comments below:
>
>
> **On results when varying $\alpha$**
>
> We apologize that the discussion of $\alpha$ was brief in the paper. In Section 5.2, we mention that we run an initial hyperparameter sweep for $\alpha$ over $[ 0,0.2, 0.4,0.6,0.8,1]$ and select the best alpha which is 0.6. The results presented in 5.2 include $\alpha= 0.6$. We reproduce Table 2 with varying value of \alpha below separately for CKA and Procrustes:
>
> **CKA**
>
> | $\alpha$ | RTE  | MRPC      | COLA  |
> | ----- | ---- | --------- | ----- |
> | 0     | 64.6 | 81.1/87.3 | 52.0    |
> | 0.2   | 64.2 | 81.8/87.7 | 51.8  |
> | 0.4   | 64.6 | 81.8/87.8 | 51.8  |
> | 0.6   | 64.6 | 81.3/84.6 | 52.0    |
> | 0.8   | 64.6 | 80/86.9   | 51.7  |
> | 1     | 47.3 | 0.68/0.81 | 13.58 |
>
>
> **Procrustes**
>
> | $\alpha$ | RTE  | MRPC      | COLA |
> | ----- | ---- | --------- | ---- |
> | 0     | 63.4 | 79.1/85.9 | 51.0   |
> | 0.2   | 64.2 | 81.3/87.3 | 54.7 |
> | 0.4   | 64.6 | 83.3/88.6 | 54.2 |
> | 0.6   | 65.7 | 83.5/88.7 | 55.0   |
> | 0.8   | 65.1 | 82.8/85.5 | 47.8 |
> | 1     | 58.8 | 68/81.2   | 19.8 |
>
>
> In general, we do not find the best performance with \alpha=1; in general excluding KD entirely leads to the worse performance. In fact, this is true for both CKA and Procrustes, which indicates that, as the reviewer pointed out, the performance might be less about better/worse feature structural alignment. We generally find that this aligns with the results we presented in Section 5.2. We do not claim that better aligning geometric structure has a causal link with better downstream performance; but that integrating better aligning metrics can lead to a better performance when combined with KD and cross entropy.

---

> ### Comment · Reviewer_p4dj · 2025-11-21
> **Rebuttal Response 1**
>
> **R1: Causal link between feature geometry/LRH and distillation performance:**
>
> I find the author's response to this question poor. If there is no causal link between LRH and distillation performance, then it feels as though the synthetic experiments are intentionally misleading, and the theoretical explanation, while complete, does not explain what occurs in practice. It is much more likely that Procrustes introduces a form of regularisation that is detached from orthogonal alignment. Additionally, as reviewer 66UJ has highlighted, the use of Procrustes distance has been used in KD. I find the novelty of this work to be severely lacking, and the explanation for misalignments between theoretical results and empirical findings remains inadequate and requires further clarification.
>
> **R2: Clarifying the baseline results:**
>
> It is highly unsatisfactory to purely copy results from a paper and not to reproduce the results and then include these results in your paper without stating this explicitly. Such malpractice can lead to error propagation and is a poor research practice. Furthermore, it is very misleading to present these results as though they were not copied from other texts. It leads reviewers to overestimate your empirical evaluation. This practice risks setting misguided future baselines, which can misinform follow-up studies, perpetuating inaccuracies that undermine the integrity of scientific progress.
>
> **R3: Statistical significance presented in the results:**
>
> I am unclear why the authors felt it necessary to clarity that Table 3 does follow standard empirical practice - the review already acknowledges this. The critique is why Tables 1 and 2 do not follow this practice. It is simply incorrect to state that classification tasks do not require error bars. All results require error bars to understand the distribution of the data. Also, if the classification tasks were not run across multiple random seeds, then the statistical significance is incorrect, as one cannot have reliable statistical significance when comparing sample sizes of 1.
>
> **R4 Distillation scaling:**
>
> Without conducting experiments over a range of student scales, it is hard to verify the limits of transfer and its interaction with capacity. I suggest implementing a minimal yet revealing capacity sweep by evaluating  a range of student sizes to efficiently expose scaling limits.
>
> **R5 On results when varying 𝛼:**
>
> Thank you for providing this analysis. However, this analysis raises some questions which underscore some of my main concerns with the empirical rigour of the work.
>
> 1. Why is there no standard deviation reported?
>
> 2. Given that as alpha increases, the weighting on the D_sim term increases, why, when alpha is 0, are the results different? They should be the same if you are doing a like-for-like comparison.
>
> 3. If this is not an apples-for-apples comparison, it makes it hard to trust the reliability of the results.
>
> 4. Please report all results to the same number of decimal places - this goes for the tables in the main paper, too. Standardise results reporting makes finding easier to process and increases the reliability of findings.
>
> 5. The results presented in Table 2 are biased towards showing higher results for Procrustes. At alpha 0.6, the results are best for Procrustes, but for CKA, they are best at alpha 0.4, which shows a closer gap between CKA and Procrustes.
>
> Unfortunately, this small table of results largely encapsulates my main concerns for the empirical rigour in this work. Provided that this work attempts to benchmark the Procrustes metric, I have serious concerns about the empirical evaluation that this work conducts.
>
> **Overall Comments:**
>
> Provided this paper is benchmarking a new method, ensuring that the evaluations are conducted properly and correctly is a core part. Furthermore, this reduces my confidence in the authors' having conducted robust and meaningful comparisons.
>
> Also, only providing the computational comparison of costs during the rebuttal phase feels like another empirical mishap of this paper. Provided that, in response to reviewer Xfq5, it is revealed that this method is circa 3x more expensive than using CKA, I find it hard to believe that this method will be used in practice due to its highly marginal gains in accuracy. To elaborate, the Procrustes method (which has questions surrounding novelty) requires in excess of 300% more compute resources while only delivering marginal improvement in performance. As a result, the computational trade-offs of using Procrustes for KD in practice is unfavourable.
>
> At this moment, I am inclined to decrease my score and increase my confidence in light of your responses to this review and the other reviews. At present, I recommend a strong reject of this paper, as I do not have confidence that the empirical rigour of this work can be improved within the time provided by the rebuttal.

---

### Official Review · Reviewer_66UJ · 2025-10-30

**Soundness:** 2
**Presentation:** 3
**Contribution:** 2
**Rating:** 4
**Confidence:** 4

**Summary:**

This paper is about feature-based knowledge distillation for LLMs. It shows that the current methods fail to preserve the teacher's geometric structure and proposes a theoretically grounded, geometry-aware distillation based on Procrustes distance and the Frobenius norm of the feature gram matrix. These findings are supported experimentally and compared to the current baselines.

**Strengths:**

The main idea is presented and discussed clearly. The paper is well-organized, and the details of the experiments and findings are well presented.

**Weaknesses:**

The main weakness of this work is the limited scope, which does not include multi-teacher and or multi-modal distillation and VLMs. Also, the use of Procrustes for KD already exists in the literature. So the novelty of this work is not well discussed.

**Questions:**

- In both Theorems 1 and 2, it is assumed that $R_t$ and $R_s$ are centered, unit-norm matrices. Can authors discuss how these assumptions are realistic (can be explored experimentally) and what the possible impacts of not respecting these assumptions are on the experimental results?
- Looking at Table 2, how does the poor performance of "Procrustes + KD " compare with "CKA + KD " align with the (theoretical) finding of the paper?
- Theoretically, can this method be applied to the multi-teacher KD? Can this be done by some experiment?

---

> ### Author Response · Authors · 2025-11-20
> **response (1/2)**
>
> We thank the reviewer for their feedback and valuable insight. We address their concerns below:
>
> **On the limited scope**
>
> We thank the reviewer for connecting our work and identifying points where we can expand upon the work. We respectfully clarify that multi-teacher and multimodal distillation falls outside the scope of the main research question we are aiming to address in the paper. Our core contribution is not to provide a flexible distillation method for all architectures, but to show that, assuming the linear representation hypothesis in language models holds true, methods for distilling features such as CKA are not sufficient to capture the geometrical structure present in the features. We believe that this problem is independent of model architectures or modalities. We focus on the models and datasets in the paper precisely because metrics like CKA have been previously applied and shown to have a performance improvement when applied to them.
>
> **Novelty with respect to Procrustes**
> > Also, the use of Procrustes for KD already exists in the literature. So the novelty of this work is not well discussed
>
> While we agree with the reviewer that while Procrustes is a classical metric that is used across various tasks in machine learning, we were unable to find previous works that have used it for distillation in language models. If the reviewer were to point us to papers that have used Procrustes in distillation, we would be happy to include them as references in the paper. Regardless, we have expanded Section 2 in the paper to include some works, primarily in vision models, that have imposed a form of geometric loss for distillation, some of which have partial connection with Procrustes.
>
> Furthermore, even if Procrustes has been pointed as a distillation method, we believe our theoretical analysis showing the equivalence between minimizing the Procrustes loss and the Feature Gram matrix (Theorem 3) is a novel contribution.
>
>
> **Centering and unit norm assumptions**
> >In both Theorems 1 and 2, it is assumed that $R_s$ and $R_t$ are centered, unit-norm matrices. Can authors discuss how these assumptions are realistic (can be explored experimentally) and what the possible impacts of not respecting these assumptions are on the experimental results?
>
>  We note that centering is a standard pre-processing step when computing both CKA and Procrustes. We center the representations in a batch before computing the value of the metric in our experiments in practice.
>
> Representations are usually normalized to have unit norm to ensure that Procrustes distance exhibits invariance to isotropic scaling (i.e for any $\lambda >0, D_P (X,Y)  = D_P (\lambda X , Y)$ ). We impose unit norm restrictions during our theoretical analysis for a fair comparison. Note that CKA is invariant to any isotropic scaling, and thus without making $D_P$ and $D_{FG}$ also invariant to isotropic scaling, the theoretical results we prove are trivially true. In practice, when computing $D_P$ and $D_{FG}$, we normalize representations and face no issues with computation or backpropagation.
>
>
> **Multi-teacher scenarios**
> >Theoretically, can this method be applied to the multi-teacher KD? Can this be done by some experiment?
>
> All the loss functions we have introduced in the paper could be used for multi-teacher scenarios. Since all of Procrustes, CKA and the feature gram matrix work on pairwise relationships between features, the loss function can be extended to any number of teachers. However, multi-teacher setups might include additional design choices that are unrelated to feature geometry. Furthermore, we are not aware of other works that have used CKA or projection losses alongside a multi-teacher setup in language models. As such, we believe the fundamental question of whether matching hidden states in multi-teacher setup is an open question.

---

> > ### Comment · Reviewer_66UJ · 2025-11-22
> >
> > I appreciate authors for their responses.
> >
> > Regarding the $\textbf{Limited scope}$, respectfully, I do not agree that the multi-teacher or multi-modal cases are out of the scope of this work, otherwise the title is misleading. Studying the proposed work in its general form where there are more than a teacher each (or a subset) with different modalities are exactly aligned with both practice and the way this draft is presented.
> >
> > Regarding the $\textbf{Novelty}$, there are some works used Procruste for KD that can be found with a quick search (like "Geometric Knowledge Distillation via Procrustes Analysis for Efficient Motion Sequence Classification" by Geometric Knowledge Distillation via Procrustes Analysis for Efficient Motion Sequence Classification by B.De et al.,) and also other works used Gram-matrix for KD. So the use of such distance for KD does not seem to be novel however, your work provide a more solid theoretical way to show why it can be useful for KD. I would suggest to name those related work in the draft.
> >
> > Regarding the $\textbf{Centering and unit norm assumptions}$, thanks for the clarification.
> >
> > Regarding the $\textbf{Multi-teacher scenarios}$, thanks for the response, but my question regarding that remained unanswered!

---

> ### Author Response · Authors · 2025-11-20
> **response (2/2)**
>
> **On the performance of Procrustes**
> >Looking at Table 2, how does the poor performance of "Procrustes + KD " compare with "CKA + KD " align with the (theoretical) finding of the paper?
>
> We thank the reviewer for this observation which allows us to delineate the theoretical and experimental sections of the paper. While we theoretically show that Procrustes captures the feature structure much better than CKA, feature structure is, by itself, not causally linked with improved performance. As we also mention to reviewer 7wxQ above, it is likely that the stronger geometric constraint imposed by Procrustes can only be fully realized once high quality downstream signals are provided. KD includes soft labels from the teacher, which is itself noisy. When fully supervised labels are given, Procrustes+KD+FT always tends to do better.
>
>
> We sincerely thank the reviewers for their thoughtful and constructive feedback. These comments have helped us clarify our contributions and have prompted several directions we will continue to think about moving forward. We hope the revisions satisfactorily resolve the issues raised and are reflected in the reviewers’ updated assessment.

---

> > ### Comment · Reviewer_66UJ · 2025-11-22
> >
> > Regarding the $\textbf{Performance of Procrustes + KD}$, thanks for the clarification. So then it open the question of the complexity of adding Procrustes to the KD+FT, as looking at the Table 2, the gain is not significant. Could you please discuss the complexity regarding adding Procrustes?
> >
> > At this point, I keep my score as before and I wait to hear more from the authors regarding my responses.

---

### Official Review · Reviewer_7wxQ · 2025-10-31

**Soundness:** 2
**Presentation:** 2
**Contribution:** 2
**Rating:** 4
**Confidence:** 4

**Summary:**

This work tackles the conflict issues generated in graph manipulation steps of generated scene graphs for downstream tasks.

**Strengths:**

1. Fundamental issue raising for the use of distance metric on knowledge distillation
2. Theoretical works to find the problem of CKA, and conditions for optimal correlation.

**Weaknesses:**

### 1. Unclear Problem Definition

This work proposes using the FG matrix or Procrustes distance instead of CKA. However, the problem setup presents that CKA performs worse than these metrics, not suggesting a more detailed problem of CKA. As a result, it is difficult to understand the actual problem being solved and benefits by replacing CKA with these alternatives. This should be more clearly elaborated.

### 2. Unclear Benefit of Using the Proposed Two Metrics
As a consequence of the first issue, it is not clearly shown which structures or representations can be better aligned in knowledge distillation compared to CKA, since the specific problem with CKA in aligning structures is not clearly defined. Furthermore, the validation section lacks qualitative analysis addressing this issue.


### 3. Missing Issue in Validation
A question comes up: "Does the number of orthogonal vectors indicate better feature structure alignment?"
While I agree with the intuition behind the results of Theorem 2, it remains unclear which structures are more accurately transferred, and whether the orthogonal vector size indicates effectively the degree of structure alignment.

a) The results only show the impact of reduced expressiveness due to a low number of orthogonal vectors. We do not know whether this is a dominating cause, especially compared to the distance metric’s potential bias in aligning feature structures. It should be shown that the orthogonal vector size meaningfully impacts alignment beyond the quantitative performance evaluation.

b) Even if orthogonal vector size is an important indicator of alignment quality, student feature representations may have fewer dimensions to represent the diversity of features (intrinsic dimension), which could require a lower number of required vectors. Learning unnecessarily large orthogonal vectors may therefore be unnecessary.

**Questions:**

### 1. Why is the performance of Procrustes + KD worse than CKA + KD in Table 2?
It requires more justification why using orthogonal vector size does not appear positively.

### 2. Large-Scale Problems Are Not Evaluated in Table 1 and 2. Why?
It is expected that the proposed measure should work for larger-scale problems if it truly enables more accurate alignment of feature structures. However, Table 2 suggests that more accurate feature representation alignment does not necessarily benefit the preservation of some student model structures in general KD settings. This raises the question: do we really need a more precise distance measure for knowledge transfer, or should we focus on selectively transferring features to avoid?

### Minor comment on Demonstration
The figures on page 7 have no captions or figure indices.

---

> ### Author Response · Authors · 2025-11-20
> **response  (1/2)**
>
> We thank the reviewer for their engagement and feedback. We address the reviewer’s concerns below:
>
> **The problem definition and inadequacies of CKA**
>
> > However, the problem setup presents that CKA performs worse than these metrics, not suggesting a more detailed problem of CKA. As a result, it is difficult to understand the actual problem being solved and benefits by replacing CKA with these alternatives. This should be more clearly elaborated.
>
> We thank the reviewer for raising this issue and hope we can better articulate the theoretical results that explicitly mentions the specific problem with CKA. In particular, Theorem 1 presents the case where CKA can have a very small value, but the inner product structure (as defined by the Frobenius norm of the difference in Gram Matrix)  in the representations are vastly different. We focus on an orthogonal geometry in the synthetic experiments because (i) The linear representation hypothesis and the superposition principle means that language model features are likely to have an approximately orthogonal structure and (ii) When features are orthogonal the $||R_t R_t^T - J_n||_2$  in Theorem 1 is maximized. Thus, to be precise, we can explicitly say that CKA does worse in alignment whenever the Gram matrix of the features deviates from the all ones matrix ($J_n$) the most. This is maximal when all features are orthogonal to each other.
>
> **Number of orthogonal vectors as performance quality**
> > A question comes up: "Does the number of orthogonal vectors indicate better feature structure alignment?" While I agree with the intuition behind the results of Theorem 2, it remains unclear which structures are more accurately transferred, and whether the orthogonal vector size indicates effectively the degree of structure alignment.
>
> > The results only show the impact of reduced expressiveness due to a low number of orthogonal vectors.
>
> We want to clarify that the number of $\epsilon$-orthogonal vectors is only used as a quantitative measure during synthetic experiments, not distillation in section 5.2 and 5.3. **Our main claim in the paper is not that increasing the number of orthogonal vectors in language models hidden states increases distillation performance, but that using loss functions that better preserve inner product geometries (regardless of whether the geometry is orthogonal or not) for hidden state matching tend to do better.** Note that both Procrustes and Feature Gram and not just maximizing the number of orthogonal features, but implicitly also aligning the inner product geometry between student and feature representations. We do agree with the reviewer that the number of orthogonal features, or just geometry is not causally linked with improved performance; in fact in section 5.2, we explicitly show that just aligning just the feature structure does not lead to an increase in performance. However, **our results do show that, when paired with KL divergence and cross entropy loss, better feature alignment can lead to better performance.**
>
> **Instrinsic dimension of the student**
> > student feature representations may have fewer dimensions to represent the diversity of features (intrinsic dimension), which could require a lower number of required vectors
>
> We want to re-iterate that we are not explicitly enforcing orthogonality when using our loss functions, but instead better aligning the geometric structure. Furthermore, the reviewer’s points about the intrinsic dimensionality of the representations is valid if we were working with exact orthogonality (i.e $<x,y>=0$). Since $d_s < d_t$, the number of orthogonal vectors in $d_s$ will always be lower than the one in $d_t$. This is exactly why we are working with $\epsilon$-orthogonality (i.e $<x,y> \leq \epsilon$), since the Johnson–Lindenstrauss lemma [1] guarantees that there are at least $2^{O(\epsilon^2 d_s)}$ vectors in $d_s$ dimensional space that are $\epsilon$-orthogonal to each other. Thus, as long as (i) teacher features lie in a $d_s$ dimensional subspace or (ii) the number of features are less than $2^{\epsilon^2 d_s}$ the students can be better aligned. We thank the reviewer for pointing this abiguity out and note that we have included a brief note of this in the paper (Lines 136-142)
>
>
> [1] Dasgupta, Sanjoy, and Anupam Gupta. "An elementary proof of a theorem of Johnson and Lindenstrauss." Random Structures & Algorithms 22.1 (2003): 60-65.

---

> ### Author Response · Authors · 2025-11-20
> **response (2/2)**
>
> > 1. Why is the performance of Procrustes + KD worse than CKA + KD in Table 2?
>
> We anticipate this is a result of the downstream signal (the classification task) not being fully utilized. Without FT, matching the student embeddings to the outputs only depends upon (noisy) soft labels from the teacher. Procrustes imposes stronger geometric constraints, however this geometric constraint cannot, by itself, improve downstream performance. When combined with supervised labels through fine-tuning, we do find that Procrustes does better than CKA.
>
>
> > It is expected that the proposed measure should work for larger-scale problems if it truly enables more accurate alignment of feature structures. However, Table 2 suggests that more accurate feature representation alignment does not necessarily benefit the preservation of some student model structures in general KD settings. This raises the question: do we really need a more precise distance measure for knowledge transfer, or should we focus on selectively transferring features to avoid?
>
> We agree with the reviewer that actively selecting better features can help distillation performance. However, we also believe that having a precise understanding of the strength and deficiencies of loss function can help improve the performance. Our theoretical framework allows this precise understanding which can be integrated into any form of feature transfer.
>
>
> We thank the reviewers for their thoughtful and constructive feedback. We have carefully addressed each comment and revised the manuscript to clarify the novelty, theoretical contributions, and empirical strengths of our work. We believe the updated version fully resolves the concerns raised, and we would be happy to clarify any other concerns the reviewer may have.

---

### Official Review · Reviewer_Xfq5 · 2025-10-31

**Soundness:** 3
**Presentation:** 3
**Contribution:** 3
**Rating:** 6
**Confidence:** 3

**Summary:**

The paper proposes a new approach for knowledge distillation from large teacher models to smaller student models by explicitly applying Procrustes distance and the Frobenius norm of Feature Gram Matrix as the objective. The paper theoretically demonstrates that existing knowledge distillation objectives, such as projection-based mean squared loss or Centered Kernel Alignment (CKA), cannot capture the feature structure even when optimized perfectly (zero loss). And the proposed Procrustes distance-based objective is theoretically and practically better. The authors provide comprehensive theoretical proof and conduct knowledge distillation experiments on various tasks to show that the proposed objective outperforms existing methods on distilling encoder-only and decoder-only language models.

**Strengths:**

1. Strong theoretical motivation showing why existing feature-based losses (e.g., CKA, projection MSE) fail to preserve true geometry.

2. Novel geometry-aware losses (Procrustes and Gram-matrix) that align representational structure up to orthogonal transformations.

3. Empirical validation across encoder (BERT) and decoder (OPT) models, covering both classification and instruction-following tasks.

**Weaknesses:**

1. Lack of experiments to show if the observations generally hold over multiple sizes of the student models. It would be great if the authors could add analysis, even only on one task, to show the performance on student models with different intermediate sizes.
2. There are also works on using mutual information maximization as a knowledge distillation objective. It would be better to discuss this branch of works either theoretically or practically.

For other concerns, please refer to the questions below.

**Questions:**

1. On line 90-93, it might be a bit confusing what a “mode” is referring to, especially for people who are not familiar with this field. Maybe add one more sentence here to explain it.

2. Do you have varied analysis on the choices of student dimension for the experiments in section 5.1 to check if the observations are consistent over teacher-student dimension differences?

3. In figure 2(b), there are spikes of CKA when optimizing over Procrustes, any explanation for this?

4. It seems that for decoder-only models, the performance difference between Procrustes and CKA (or FG) is smaller, do you have any intuitions why?

5. It would be great if the authors could do some analysis on the convergence speed of Procrustes compared to CKA (or FG) for section 5.2 and 5.3

6. What's the computation overhead of the proposed objective compared to CKA?

---

> ### Author Response · Authors · 2025-11-20
> **response (1/2)**
>
> We thank the reviewer for their thoughtful and constructive feedback, and for drawing connections from our work to contemporary work in distillation. We are also happy that the reviewer found strength in our theoretical analysis and motivations. We addressed their concerns below:
>
> **Experiments with different hidden sizes**
>
> > Lack of experiments to show if the observations generally hold over multiple sizes of the student models. It would be great if the authors could add analysis, even only on one task, to show the performance on student models with different intermediate sizes
>
> >Do you have varied analysis on the choices of student dimension for the experiments in section 5.1 to check if the observations are consistent over teacher-student dimension differences?
>
> We have added synthetic experiments that complement the main one presented in the paper and have included them in the appendix. (B.5) We choose $d_t = 5120$ and $d_s =2560$ ( the same hidden dimensionality scenario as the OPT models) and $d_t = 1024$ and $d_s = 768$ (the same hidden dimensionality as the BERT models). We find that observations are largely the same, but do note that the number of $\epsilon$-orthogonal vectors fluctuate more in OPT-scale hidden sizes. However, even in that case, we find that Procrustes and FG tend to better capture the teacher’s orthogonal geometry, while both CKA and linear projection collapse.
>
> **Links with mutual information**
>
> >There are also works on using mutual information maximization as a knowledge distillation objective. It would be better to discuss this branch of works either theoretically or practically.
>
> We thank the reviewer for connecting our work to mutual information. We believe that mutual information based distillation is an orthogonal, but closely linked method compared to our geometric approach. In fact, after the reviewer's suggestion we have references that have theoretically linked variational mutual information with hidden state losses [1] . These results have shown that in general, when X and Y are representations drawn from a distribution, maximizing the variational bound of mutual information is equivalent to minimizing the L2 distance or maximizing the cosine similarity. Given the close-link between each of our loss funcitons and inner products, we beleive the same theoretical observations that [1] that have derived for cosine similarity can be carried over to our metrics. We are working through this connection and will include this in the paper after this is clear to us.
>
>
>
> **Training spikes**
> >In figure 2(b), there are spikes of CKA when optimizing over Procrustes, any explanation for this?
>
> We believe this is an artifact of noisy gradient descent. At each step, we are calculating the gradient and optimizing with respect to a batch of vectors, but the CKA is calculated over the entire dataset. So stochastic variability in the sampled minibatches can induce abrupt fluctuations in the CKA value. However, these perturbations tend to dissipate quickly as the optimization proceeds and the estimates stabilize as can be seen in Figure 2(b) and Figure (8)
>
>
> **Performance disparity in OPT**
> > It seems that for decoder-only models, the performance difference between Procrustes and CKA (or FG) is smaller, do you have any intuitions why?
>
> Both the teacher and the student decoder-only model that we study have almost 2x the hidden size compared to BERT. As a result, it is possible that the feature space for the student model is already expressive enough and the geometrical alignment that we achieve as a result of a better alignment measure does not add as much novel information to the feature space. It is also possible that the representations exhibit less approximately orthogonal-like geometries, which is where Procrustes tends to do a better job. We thank the reviewer for this observation and agree that looking into this disparity can be an interesting follow up work.
>
>
> [1] He, Haoyu, et al. "Distiller: A systematic study of model distillation methods in natural language processing." Proceedings of the Second Workshop on Simple and Efficient Natural Language Processing. 2021.

---

> ### Author Response · Authors · 2025-11-20
> **response (2/2)**
>
> **Computational overhead in Procrustes**
> >What's the computation overhead of the proposed objective compared to CKA?
>
> Procrustes involves computing an SVD, so it is computationally more expensive than CKA. We include the time taken (in seconds) for a single step of optimization through a batch on each of MRPC, COLA and RTE using a single A100 GPU and a batch size of 64 below:
>
> |      | Procrustes | CKA   |
> | ---- | ---------- | ----- |
> | COLA | 47.62      | 13.55 |
> | RTE  | 47.61      | 13.33 |
> | MRPC | 47.62      | 13.15 |
>
>
> We sincerely thank the reviewer for their thoughtful and constructive feedback. We have carefully addressed all comments and revised the manuscript accordingly. We hope these changes clarify the novelty and strengths of our contributions,

---

### Meta-Review · Area_Chair_QmU9 · 2026-01-11

**Summary:**

While most reviewers appreciate the theoretical motivation and good performance, they raised several concerns about marginal novelty, unjustified motivational hypothesis, missing experimental analyses, and limited evaluation. In the rebuttal and discussion phase, the authors failed to assuage the concerns, and some major concerns remain, in particular, the issue of novelty, the justification of the linear representation hypothesis, and the fairness of experimental comparisons. Given the prior work that used Procrustes for KD, as pointed out by reviewer 66UJ, the novelty of the proposed method is unclear. Furthermore, the discussion with reviewer p4dj reveals that the motivation for the linear representation hypothesis is inconsistent across the manuscript, fails to support the distillation performance, and renders the synthetic experiments pointless; also, the experimental results in the paper are not conducted with experimental rigor in a principled way. Agreeing with reviewer p4dj’s concerns, AC recommends rejection.

**Reviewer Concerns:**

The main concerns were marginal novelty, unjustified motivational hypothesis, missing experimental analyses, and limited evaluation. Among them, only some parts of the missing experimental analyses, raised by reviewers Xfq5 and 7wxQ, are clearly addressed. The concerns about novelty, justification, and evaluations remain outstanding. Given the prior work that used Procrustes for KD, as pointed out by reviewer 66UJ, the novelty of the proposed method is unclear. The discussion with reviewer p4dj reveals that the motivation for the linear representation hypothesis is inconsistent across the manuscript, fails to support the distillation performance, and renders the synthetic experiments pointless; also, the experimental results in the paper are not conducted with experimental rigor in a principled way.

**Reviewer Scores:**

Reviewer Xfq5 would have retained the original score 6.
Reviewer 7wxQ  would have changed the score from 4 to 6.
Reviewer 66UJ  retained the original rating of 4 after discussion.
Reviewer p4dj  have changed the score from 2 to 0 after discussion.

---

### Decision · Program_Chairs · 2026-01-26

Reject